# Basin-scale impacts of hydropower development on the Mompós Depression wetlands, Colombia

Héctor Angarita[1,2], Albertus J. Wickel[3a], Jack Sieber[3b], John Chavarro[4], Javier A. Maldonado-Ocampo[5], Guido A. Herrera-R[5,6], Juliana Delgado[1], David Purkey[3]

[1] The Nature Conservancy, Bogotá, Colombia

[2] Grupo de ecología y territorio, Facultad de estudios ambientales y rurales, Pontificia Universidad Javeriana, Bogotá, Colombia

[3] Stockholm Environment Institute US Center [a]Davis, California, USA; [b]Somerville, Massachusetts, USA

[4] Centro de Investigación en Ciencias y Recursos GeoAgroAmbientales CENIGAA, Neiva, Colombia

[5] Laboratorio de Ictiología, Unidad de Ecología y Sistemática (UNESIS), Departamento de Biología, Facultad de Ciencias, Pontificia Universidad Javeriana. Bogotá, Colombia

[6] Laboratoire Evolution et Diversité Biologiques (EDB), Paul Sabatier University, Toulouse III, France

*Correspondence to*: H. Angarita (hangarita@javeriana.edu.co)

**Abstract:** A number of large hydropower dams are currently under development or in an advanced stage of planning in the
Magdalena River basin, Colombia, spelling uncertainty for the Mompós Depression wetlands, one of the largest wetland systems in South America at 3400 km$^2$. Annual large-scale inundation of floodplains and associated wetlands regulates water-, nutrient-, and sediment cycles, which in turn sustain a wealth of ecological processes and ecosystem services, including critical food supplies. In this study, we implemented an integrated approach focused on key attributes of ecologically functional floodplains: (1) hydrologic connectivity between the river and the floodplain, and between upstream and downstream sections;
(2) hydrologic variability patterns and their links to local and regional processes; and (3) the spatial scale required to sustain floodplain-associated processes and benefits, like migratory fish biodiversity. The implemented framework provides an explicit quantification of the non-linear or direct response relationship of those considerations with hydropower development. The proposed framework was used to develop a comparative analysis of potential effects of the hydropower expansion necessary to meet projected 2050 electricity requirements. As part of this study, we developed an enhancement of the Water
Evaluation and Planning system (WEAP) that allows resolution of the floodplains water balance at a medium scale (~1000 to 10 000 km$^2$) and evaluation of the potential impacts of upstream water management practices. In the case of the Mompós Depression wetlands, our results indicate that potential additional impacts of new hydropower infrastructure with respect to baseline conditions can range up to one order of magnitude between scenarios that are comparable in terms of energy capacity. Fragmentation of connectivity corridors between lowland floodplains and upstream spawning habitats and reduction of
sediment loads show the greatest impacts, with potential reductions of up to 97.6 and 80%, respectively, from pre-dam conditions. In some development scenarios, the amount of water regulated and withheld by upstream infrastructure is of similar magnitude to existing fluxes involved in the episodic inundation of the floodplain during dry years and, thus, can also induce substantial changes in floodplain seasonal dynamics of average-to-dry years in some areas of the Mompós Depression.

**Keywords**: Cumulative impacts on freshwater systems, River fragmentation, Migratory fish, Floodplains dynamics, Sediment entrapment

## 1    Introduction

Hydropower is a fundamental component of many countries' energy supply due to comparative advantages such as long-term economic efficiency, flexibility to adapt to high-frequency demand fluctuations, and greater regulation of hydrologic variability for other water users. Recently, climate change considerations have reawakened interest in hydropower development for its potential contributions to low-carbon economies and reduced dependency on fossil fuels.

Dam and reservoir construction and operations are one of the main drivers of global change in freshwater systems (Dynesius

and Nilsson, 1994; Grill et al., 2015; Zarfl et al., 2014). There are numerous examples worldwide of how changes in flow, sediment, and temperature regimes; loss of river connectivity; and other impacts associated with reservoirs and dams cumulatively affect the physical and ecological processes that determine the integrity of major river systems, and in particular, of riverine lowland floodplains and wetlands (Arias et al., 2014; Dang et al., 2016; Grill et al., 2015; Opperman et al., 2010; Tockner and Stanford, 2002).

Riverine floodplains and wetlands are ecosystems of high biodiversity and productivity (Tockner & Stanford, 2002), providing numerous benefits, including stable water supply, support for fisheries, flood risk mitigation, carbon regulation, and improved water quality (Zedler and Kercher, 2005). Floodplain systems – despite their comparatively small spatial footprint – generally exceed the productivity of purely terrestrial or purely aquatic ecosystems (Bayley, 1995; Tockner and Stanford, 2002). Due to their central role in processes operating at the basin scale, and to the economic value of the numerous services they provide,

hydrologically and ecologically functional riverine floodplains should be factored into sustainable water management infrastructure development. Such consideration should go beyond project-scale environmental impact assessments to consider the cumulative effect of all interventions located upstream (Dang et al., 2016; Fitzhugh and Vogel, 2011; Yang and Lu, 2014). Basin-scale analysis aims to explicitly take into account the benefits of water management infrastructure along with potential repercussions to long-range processes and services that freshwater systems naturally provide; such analysis is especially

relevant because the hierarchical and nested character of river networks and their associated ecosystems lead to non-linearity of impacts (Fullerton et al., 2010; Grill et al., 2015). For example, habitat fragmentation is highly dependent on the geographic configuration of artificial barriers (Fausch et al., 2002); unique disturbances at specific locations can have system-wide impacts, and multiple dams, while individually disconnecting relatively small parts of the river network, can together disconnect large portions of non-substitutable habitat, constraining key ecological or physical processes, like fish migration

from floodplains to upstream spawning habitats, or sediment and nutrient transport (López-Casas et al., 2016; Yang and Lu, 2014).

Floodplains and associated wetlands rely on longitudinal, lateral, and vertical connectivity which all affect the extent, depth, duration, and frequency of inundation. Cumulative flow alteration associated with upstream reservoir operation disrupts these

hydrologic processes, which, in turn, affect multiple physical and ecosystem characteristics and processes, like floodplain topography; deposition of nutrients and organic matter in the floodplain; recharge of the water table; recruitment, dispersion, and colonization of plants; fish migration triggers; and access to soil moisture, among many others (Arias et al., 2014; Poff and Zimmerman, 2010).

Like habitat fragmentation, changes in the magnitude, frequency, duration, and timing of river flows also exhibit non-linear cumulative behavior (Dang et al., 2016; Poff et al., 1997; Richter et al., 1998). While the artificial regulation effect of individual dams on hydrologic alteration depends both on reservoir storage capacity in comparison to the natural river discharge and on the operational rules (Williams and Wolman, 1984), at the basin level, dam placement determines both the spatial extent and the degree of alteration; certain relative dam locations can enable (or preclude) attenuation of streamflow from tributary rivers,

and multiple dams located in the same river branch or sub-basin can amplify artificial regulation – resulting in hydrologic alteration greater than the sum of the individual effects of single reservoirs and propagating impacts hundreds or thousands of kilometers downstream (Angarita et al., 2013; Fitzhugh and Vogel, 2011; Piman et al., 2016; Richter et al., 1998).

The decrease in sediment loading due to reservoir trapping is another important driver of change in freshwater systems (Vörösmarty et al., 2003). Deficits in sediment loads are responsible for a number of impacts, like erosion and subsidence of

river deltas (Syvitski et al., 2009), progressive incision or incremental changes in channel sinuosity and bank erosion (Grant et al., 2003), and transformation of wetlands and floodplains into permanent water bodies; and indirectly, as consequences of these impacts, de-stabilization of infrastructure like bridges, bank protections, levees, etc.

Medium and large hydropower plants in the Magdalena River basin (MRB) with a total capacity of 6.89 GW currently supply 49% of the electricity consumed in Colombia. Faced with growing demand – by 2050, electricity use in Colombia is expected

to increase by between 105% and 147% with respect to 2010 (UPME, 2015) – there is great interest in further developing the remaining MRB hydroelectric potential, estimated at ~35 GW (Departamento Nacional de Planeación, 1979). Due to its proximity to existing transmission infrastructure and to urban areas that represent 75% of the energy demand of the country, the Magdalena River and its tributaries make an attractive target for further hydropower expansion. Recently, basin-level impacts of MRB hydropower have been discussed in terms of a) cumulative hydrologic alteration (Angarita et al., 2013); b)

loss of longitudinal connectivity (Opperman et al., 2015); c) contribution to changes in fish productivity, extinction risk, species distribution, community composition, and extent of spawning habitat (Carvajal-Quintero et al., 2017; Jiménez-Segura et al., 2014; Pareja-Carmona and Ospina-Pabón, 2014); and d) reproductive biology of fish of economic importance (López-Casas et al., 2014, 2016; Villa-Navarro et al., 2014).

However, none of the above-mentioned studies included an integrated basin-level analysis of cumulative impacts on lowland

riverine floodplains in the MRB. In this paper, we present an assessment of the current and potential basin-scale impacts of hydropower expansion on these floodplains – the Mompós Depression wetlands. We propose an integrated framework that takes into consideration basin-level and local factors to assess system alteration. From a basin-level perspective, we first developed an integrated analysis of three main factors associated with cumulative impacts of hydropower infrastructure: 1) flow regime alteration, 2) sediment trapping, and 3) connectivity losses with upper tributaries, with an emphasis on

migratory fish species. Second, to estimate long-range hydrologic dynamics of floodplains, we developed an enhancement of the Water Evaluation and Planning system, or WEAP (SEI, 1992-2017), capable of reproducing floodplain fluxes and storage, to resolve the Mompós Depression floodplains' water balance at a medium scale (~1000 to 10 000 km$^2$) and evaluate its relationship to upstream and local water management.

## 2    System description

The Magdalena River is located in the Northern Andes Mountains and drains a biodiverse mosaic of ecosystems along its journey northward to the Caribbean Sea. The basin covers nearly one quarter of Colombia's national territory, providing sustenance, and acting as an economic and cultural life-force, especially for the more than 35 million Colombians – 70% of the country's population – who live within its bounds (Figure 1).

With a length of 1540 km, the main stem of the Magdalena is the principal riverine trade artery of the country and the main connection to the Atlantic Ocean (ARCADIS Nederland BV and JESYCA S.A.S., 2015). Following the Strahler system of stream order classification (Strahler, 1957) the MRB network ranges from small mountain tributaries (order 1), to the Magdalena at its mouth in the Caribbean Sea (order 8). The total network comprises a total length of approximately 101 109 km, of which 11 997 are medium to large rivers (Strahler order ≥4). Average flows range from $46 \pm 30$ m$^3$/s (order 4 rivers) to $7359 \pm 203$ m$^3$/s (order 8).

The Magdalena River flows between the Eastern and Central Cordillera of the Northern Andes. Tanner (1974) argued that the Magdalena River valley is an "incomplete flood plain", a term he defined in a submission by the same name. Floodplain incompleteness, according to Tanner, can result either from rapid changes in sea level or from continued tectonic deformation, the latter being a likely explanation in this intermontane basin within the active Andes orogenous zone. Incomplete floodplains are characterized by lakes, marshes, wetlands, and swamps – depressions inundated by a high water table – and lack signs of prior meandering or channel migration. Near the town of El Banco (23.5 masl), situated just upstream of what is considered the lower Magdalena, the Magdalena River is joined by the Cesar River. Downstream of El Banco, it splits into numerous channels, and is joined by two more tributaries: the Cauca and the smaller San Jorge (Figure 1). The tectonically active foreland basin of the lower Magdalena "consists of vertically accreting, levee-confined channels and adjacent extensive [Mompós] wetlands, which are interpreted as an anastomosing river sedimentary system" (Smith, 1986, p. 177). A notable feature of this basin is extensive and water-logged negative-relief elements (Lewin and Ashworth, 2014). The wetlands, dissected by numerous tie-channels, together with the permanent and temporary lentic waterbodies called "ciénagas" encompass approximately 3400 km$^2$, comprising one of the largest wetland systems in South America. About 200 km from the Caribbean Sea, downstream of the city of Mangangué (10 masl), the numerous braids of the Magdalena converge and meander as a single channel until the Magdalena splits again at Calamar, with part of the flow diverted westward to Cartagena through an altered channel system that serves as a navigation canal and part flowing into a 100 km long delta, while the main river continues to its mouth in Barranquilla.

The Magdalena is among the rivers with the highest sediment yields in South America: 560 t/km²/year – a rate approximately three times that in the Amazon, La Plata, and Orinoco rivers (Restrepo, Kjerfve, Hermelin, & Restrepo, 2006). The most recent estimate of annual sediment flux (suspended sediments) of the Magdalena is $142.6 \times 10^9$ kg yr$^{-1}$ (Restrepo, Ortíz, Otero, & Ospino, 2015). High rates of sediment transport have shaped the basin-scale morphologic and hydrologic dynamics that

determine the complex storage and exchange patterns of water in the river and adjacent plains (Posada G. and Rhenals, 2006). The discharge pattern of the Magdalena to the Mompós Depression is largely determined by the Inter Tropical Convergence Zone (ITCZ), which annually oscillates from the equator to the northern Andes and back, resulting in two rainy seasons: April–May and September–November (Poveda et al., 2011). This weather pattern typically results in predictable bimodal discharge peaks in April–May and October–November (Poveda et al., 2001; Smith, 1986). The roles of topography, soil-atmosphere

interactions, the Atlantic Ocean, and the Amazon also influence temporal and spatial rainfall patterns, resulting in the bimodal character not being equally strong across the basin (Poveda et al., 2011). The lower basin near the Caribbean coast – including the Mompós Depression – is often suggested to be unimodal in character, and the southeastern portion of the basin (approximately below 2° N) is characterized by a distinct unimodal pattern, with a June-to-August wet season (IDEAM, 2014). During intense El Niño–Southern Oscillation (ENSO) events, the ITCZ can extend anomalously far south, bringing drought

conditions to the MRB. In contrast, during La Niña events the MRB experiences heavier than normal rains and colder conditions that often extend – sometimes even bridging ITCZ events, leading to rainy periods that can last a year or longer (Poveda et al., 2001; Poveda and Mesa, 1997). The strong relation between anomalously high or low stream flow conditions at four stations in the MRB and the Oceanic Niño Index, a measure of ENSO, is illustrated in Figure 2. Observed climate variability in the MRB also exhibits oscillations at decadal or interdecadal timescales, represented by multiple macroclimatic

oscillations including Pacific Decadal Oscillation and Atlantic Multidecadal Oscillation (IDEAM, 2014).
The hydrologic variation of the lower Magdalena River and its resulting hydroperiod in the Mompós wetland system are crucial to the system's high ecological complexity and species diversity. The wetland ecosystems depend on seasonal inundation and the nutrients and sediment delivered by floodwaters. The system contains more than 226 native fish species with 129 (57%) endemic (Maldonado-Ocampo et al., 2008), and at least 16 that undertake reproductive migration from the low floodplain to

25 the foothills of the Andes (López-Casas et al., 2016). This richness and high species endemism, in addition to the proximity to main human settlements, has made the river the country's main and most productive fishery, one based on at least 40 species (FAO, 2015). Fish are the main source of dietary protein for many MRB communities (Galvis and Mojica, 2007; Lasso et al., 2011). Additionally, the wetlands and lagoons of the lower Magdalena are critical stopovers for migrating and wintering birds along the Pacific Americas Flyway, where episodic inundation is critical to fish and bird reproduction, while low-flow

conditions are important for reptile reproduction, propagation of riparian flora, and nutrient and organic matter storage (Jaramillo et al., 2015).

## 3    Data and methods

### 3.1    Basin-level considerations

#### 3.1.1    Defining dam sets for current and potential development

This study focused on existing and proposed medium and large hydropower projects, including reservoirs and run-of-river
plants. Such projects can reduce river network connectivity or produce downstream alterations. Currently the MRB upstream
of the Mompós Depression provides 70% of Colombia's hydropower, equivalent to 49% of the country's electricity supply
(XM, 2014). Ninety-five percent of the capacity is distributed over 35 plants (32 in operation and 3 expected to be completed
in 2018), with an aggregate installed capacity of 6.89 GW (and expected expansion to 9.35 GW in 2018) and 17.2 billion $m^3$
of storage (equivalent to 8.4% of the basin's average annual runoff). The remaining 5% corresponds to small hydro plants (<20
MW).

In Colombia there is no centralized or coordinated planning for hydropower site identification; expansion occurs on an
individual project basis in response to rolling auctions issued by the government based on 5- to 15-year projected needs of
additional generation capacity (Cramton and Stoft, 2007; UPME, 2012).

To account for this uncertainty, our analysis first identified and compared a set of 1000 possible future scenarios – starting
from a baseline condition that includes existing and under-construction dams – using a catalog of 97 potential project sites
identified in Colombia's 1979 hydropower inventory (Departamento Nacional de Planeación, 1979) (Table 1), considered to
be reliable by government and developers (See locations in Figure 1). We evaluated each scenario with respect to the
cumulative basin-level impacts of a) loss of river network longitudinal connectivity between wetlands and upstream tributaries,
b) boundary conditions of flow regime alteration, and c) loss of sediment input due to reservoir entrapment. Based on results
from the first 1000 scenarios, four additional sets of 100 scenarios each were generated by applying some restrictions to the
potential sites, to avoid one or more criteria – projects located on Mompós Depression tributaries (order 4+) not yet affected
by artificial barriers, mainstem projects upstream of existing projects, projects that would inundate areas with >300 inhabitants,
and projects that would inundate productive lands >300 hectares. In the context of Colombia's regulatory framework for energy
expansion, this second set of scenarios demonstrates some examples of the sensitivity of developable hydropower potential to
basin-level policy for site identification guidelines.

From the subset of scenarios that meet projected hydropower expansion by year 2050 – an equivalent hydropower capacity of
15.25±0.5 GW, or +125% with respect to 2010 (UPME, 2015) – we selected five scenarios representative of the range of
impacts and trade-offs in the basin on which to perform a more detailed analysis of the potential changes in streamflow regime
and hydrologic dynamics of the Mompós Depression wetlands. This analysis consists of a 33 yr simulation of reservoir
operations (using as reference the period 1981 to 2013). The simulation results allowed us to estimate the potential changes in
streamflow regime and hydrologic dynamics of the associated Mompós Depression wetlands.

### 3.1.2 Flow regime alteration

As part of this study, we developed a new indicator, named the *weighted degree of regulation (DOR_w)*, to perform a comparative analysis of potential cumulative impacts of the natural flow regime of multiple reservoirs at the level of an entire river basin. The indicator is based on the original DOR, applied in several regional and global assessments as a first-level approximation of flow alteration (Grill et al., 2015; Lehner et al., 2011). DOR is the fraction of a river's annual flow volume that can be withheld by reservoirs upstream of a river reach, and is calculated as the relationship between the cumulative reservoir storage upstream and the total annual river flow in a river section. Higher values indicate a greater potential alteration of the natural flow regime – particularly of seasonal patterns – due to operations effects of the reservoirs; however, the DOR indicator doesn't consider the attenuation of artificial regulation from the fraction of basin runoff not affected by reservoir operations, and as a result DOR cannot differentiate the effect of proximity of the reservoir to the interest point. In order to overcome the above limitation of the DOR, we included a weighting based on the percentage of yearly upstream runoff effectively controlled by artificial storage, or:

$$DORw_r = \frac{Q_{cr}}{Q_r} \frac{\sum_{upstream_r} V}{Q_r} \cdot 100\% \tag{1}$$

where $Q_c$ is the upstream annual runoff affected by artificial storage (m$^3$/yr), $V$ the reservoir volume (m$^3$), and $Q$ the total annual river runoff (m$^3$/yr), with the $r$ sub-index referring to specific reaches. In comparison with the previous *DOR* index, the weighting factor explicitly considers the attenuation of artificial regulation from the fraction of basin runoff not affected by reservoir operations. As *DOR_w* provides a basin-scale index of basin-level flow alteration it is thought to be particularly useful as a metric for basin-scale impacts on downstream wetland systems as found in the Mompós wetlands.

For the five selected scenarios of hydropower expansion thought to be representative of the potential range of alteration, we performed a 33 year simulation of the system to estimate boundary conditions (monthly streamflow) at the three main tributaries upstream of the Mompós Depression – the Magdalena, Cauca, and Nechí rivers – using Matlab's *ReservoirSimulator* model (Angarita et al., 2013; Ritter, 2016). This model performs a water balance of the inflows from tributary sub-basins of the reservoirs, coupled to a reservoir operations routine for hydropower production, along with other requirements such as water diversions and environmental flow obligations, when applicable (see SI-1).

For a given reservoir, the model takes into account physical and technical constraints, such as volume–elevation curve, tail-water elevation, operational levels (inactive, buffer, technical, and safety), turbine type, capacity, and efficiency. Physical characteristics for existing dams were obtained from project official documentation archives, and for projected dams from the 1979 inventory (Departamento Nacional de Planeación, 1979). MRB river topology, sub-basins, and volume–area–elevation curves were derived using the HydroSHEDS, dataset (Lehner et al., 2008; Lehner and Grill, 2013). Unimpaired flows for each sub-basin were lumped at dam sites based on observed runoff records from 1981 to 2013.

Water allocation for hydropower is based on basin-level target generation for a given time step. Target-generation for a multi-reservoir system is an extremely complex problem, subject to many interlinked factors operating at multiple time-scales, including water inflows, operational rules and technical constraints, firm energy obligations, fuel prices, and energy market

competition (Cramton and Stoft, 2007; Ritter, 2016). In order to provide a plausible estimate of the monthly variability of generation targets of hydro-plants in the MRB, we evaluated the historical monthly average plant-factor (PF; Figure 3) – the average percentage use of installed capacity – of existing medium and large hydro plants in the MRB from 2000 to 2015, based on market data (XM, 2014). Monthly average PFs for the MRB range from 41 to 85%; with most of the variation associated

with hydro-climatic oscillations, like the 2008–2011 sequence of Niña–Niño–Niña events (Figure 2). On the other hand, intra-annual monthly variation of PF in non-extreme years shows relatively stable values within a year, with a variation of 10 to 16% from dry to wet months. This is consistent with the prominent role hydropower plays in Colombian energy supply and base-load generation; cumulative storage and water allocation is able to compensate – on a monthly basis – for seasonal hydrologic variability. Based on observed PFs, we developed the following regression model to estimate average monthly PFs

for the full simulation period 1981–2013 (Adj $R^2$=0.62, Std. error=5.8%):

$$PF = -0.031 \cdot ONI + 1.205 \times 10^{-5} \cdot QL_{Calamar} + 0.233 \cdot Log\left(MA_6(QL_{calamar})\right) - 0.371 \qquad (2)$$

where *ONI* is the Oceanic Niño Index, $QL_{Calamar}$ the monthly average streamflow at Calamar (station 2903702, shown in Fig. 2), and $MA_6$ a moving average operator applied over a six month period.

### 3.1.3   Sediment trapping

We estimated basin-level entrapment or $S_e$, defined as the percentage of total sediment throughput retained by upstream reservoirs, considering two main factors: individual reservoirs' retention efficiency, and the relative locations of multiple upstream reservoirs.

To estimate trapping efficiency for each reservoir, we used Dendy's formula (Dendy, 1974). Dendy's method is a revised Brune curve, which uses an empirical expression to estimate the long-term average reservoir sediment retention efficiency

based on the ratio between capacity (*C*) and average annual inflow (*I*). A higher ratio indicates higher sediment retention efficiency, *TE*, as described by the following equation:

$$TE = 100 * \left(0.97^{0.19^{Log\left(\frac{C}{I}\right)}}\right) \qquad (3)$$

Similar to the case of flow regime alteration, relative locations of reservoirs play an important role in sediment entrapment because upstream reservoirs can significantly reduce sediment input to downstream reservoirs, and sediment yields vary across

the basin (See Restrepo et al., 2006, for a detailed analysis of the MRB). To consider the effects of relative dam locations and of sediment yield heterogeneity, we developed a routing model for reach-level sediment balance, as described by the following recursive equation:

$$SST_r = \left(\sum_{u \in Inflow_r} SST_u + E_r * A_r\right) * (1 - TE) \qquad (4)$$

where $SST_r$ is the sediment load downstream of reach *r*, *Inflow_r* the set of river reaches directly upstream of reach *r*, $A_r$ the

drainage area, and $E_r$ the contribution of sediments generated by laminar erosion and storage on the slopes, based on the Revised Universal Soil Loss Equation (RUSLE) methodology:

$$E_r = R \cdot K \cdot L_s \cdot C \cdot P \qquad (5)$$

where $E$ is laminar erosion [ton/m²/year], $R$ rain erosivity [MJ mm/m²/h], $K$ soil erodability [ton·h/MJ·mm], $L_S$ topographic factor [dimensionless], $C$ soil cover [dimensionless], and $P$ management practices [dimensionless]. Values for each of the corresponding variables were adopted from Jimenez (2016) for the MRB. Our simplified approach focuses on the primary inputs and outputs in a section of a stream according to Wilkinson et al. (2009), where the primary production process corresponds to the contribution of slope and channel erosion in the upper parts of the basin (Strahler order 1). Our main purpose was to provide a basis for comparative analyses of sediment retention in the tributary rivers of the Mompós Depression for the different hydropower expansion scenarios; therefore, we do not provide a comprehensive description of the other components of the channel sediment balance, such as sediment production by lateral migration of the channel, or bank overflow events and sediment deposition.

## 3.2    Floodplains hydrologic dynamics

We developed a conceptual hydrological model with a surface storage component that includes episodic interactions between river and wetland systems as an enhancement to the WEAP platform's existing Soil Moisture Model (SMM) (Yates et al., 2005b). The model dynamically simulates evapotranspiration, surface runoff, sub-surface runoff or interflow, and deep percolation at the sub-basin level, as well as bi-directional water transfer between river and wetland systems. The water balance is defined using a semi-distributed approach that reflects the topological relationships between basin areas or catchments, stream networks, and wetlands. The model allows for the evaluation of hydrologic dynamics associated with several factors, including alteration in the upstream flow regime, climate variability and change, and impacts of local and upstream water resource management practices, such as flood control structures and changes in connectivity between river and wetland systems.

WEAP SMM enhancements included two main modifications: the inclusion of surface storage for water balance representation at the catchment level; and the topological representation of interactions between surface storage, sub-surface storage, and the river network. WEAP's original SMM represents the water mass balance through two soil layers – the root zone and the deep zone – in lumped portions of the watershed called catchment objects, each divided into $N$ fractional areas $j$ representing different land cover types, with a water balance computed for each fractional area. The model "uses empirical functions that describe evapotranspiration, surface runoff, sub-surface runoff or interflow, and deep percolation" (Yates et al., 2005a, p.491). The modified version introduces a third storage volume (or "bucket"), corresponding to a fractional area of the catchment that accounts for surface storage. The water balance in the third bucket is determined by a) bidirectional exchanges of water (flood and return flow) with one or more sections of river and b) local inputs-outputs such as precipitation, evaporation, or percolation (Figure 4).

Water balance in the soil root zone and soil deep zone are calculated, respectively, by land cover type:

$$Sw_j \frac{dz_{1,j}}{dt} = P + Ir - PET * k_{c,j}(t)\left(\frac{5z_{1,j} - 2z_{1,j}^2}{3}\right) - (P_e + I_r)z_{1,j}^{RRF_j} - (1 - f_j)k_s z_{1,j}^2 \tag{6}$$

$$Dw_j \frac{dz_{2,j}}{dt} = (1 - f_j)k_s z_{1,j}^2 - k_d z_{2,j}^2 \tag{7}$$

The total runoff $Ro$ [volume] and baseflow $Bf$ [volume] of a given catchment are then calculated as sums of the contributions of the land cover types:

$$Ro(t) = \sum_{j=1}^{N} A_j \left[ (P + Ir)z_{1,j}^{RRF_j} + f_j\, k_s\, z_{1,j}^2 \right] \tag{8}$$

$$Bf(t) = \sum_{j=1}^{N} A_j \left[ k_d\, z_{2,j}^2 \right] \tag{9}$$

5 where

$Sw_j$ is soil root zone water storage capacity (length),

$Dw_j$, deep zone water storage capacity (length),

$z_1$, water stored in the root zone, relative to its total storage capacity (%),

$z_2$, water stored in the deep zone, relative to its total storage capacity (%),

10 $P$, precipitation and snowmelt in the catchment (length),

$Ir$, irrigation (length),

$PET$, Penman-Monteith reference crop potential evapotranspiration (length time$^{-1}$),

$k_{c,j}$, crop coefficient (dimensionless),

$f_j$, flow direction (dimensionless),

15 $k_s$, conductivity of the root zone (length time$^{-1}$),

$k_d$, conductivity of the deep zone (length time$^{-1}$),

$RRF_j$, runoff resistance factor (dimensionless), and

$A_j$, area of land cover of type $j$

Likewise, the mass balance at the floodplain and the connected river reaches ($V_{river}$), is represented by:

20 $$\frac{dV_{3,j}}{dt} = Q_l - R_l - A_3 * \left[ P_e * z_{1,j}^{RRF_j} - PET(t)(1 - k_{c,j}) \left( \frac{5z_{1,j} - 2z_{1,j}^2}{3} \right) \right] \tag{10}$$

$$\frac{dV_{river_i}}{dt} = Q_h - Q_l - Ir + R_l + Ro + Bf \tag{11}$$

where

$V_{3,j}$ is storage volume in the floodplain (volume),

$V_{river,i}$, water stored in the connected river reach (volume),

25 $A_3$, extent (area) of flooded area, given the volume of floodwater in catchment,

$Q_h$, river reach input streamflow,

$Q_l$, lateral flow between river and floodplain (volume time$^{-1}$), defined as percentage $T_f$ of the river reach streamflow, above a certain flow threshold:

$$Q_l = \begin{cases} T_f \cdot (Q_h - Q_{threshold}), & if\ Q_h > Q_{threshold} \\ 0, & if\ Q_h < Q_{threshold} \end{cases} \tag{12}$$

30 and $R_l$, return flow from floodplain to river reach (volume time$^{-1}$), defined as the percentage $T_r$ of water above a floodplain storage threshold, that flows out of the floodplain in one time step:

$$R_l = \begin{cases} T_r \cdot \left( V_3 - V_{3,threshold} \right), if \ Q_h < Q_{threshold} \\ 0 \ , if \ Q_h < Q_{threshold} \end{cases} \tag{13}$$

While there is a wide range of modeling approaches to study floodplain systems dynamics, including MIKE21 (DHI, 2016), ANUGA HMP (Roberts, 2006-2017), and HEC-RAS (USACE and RMA, 2016), conceptual approaches have several advantages, as previously discussed by Dutta et al. (2013). Our lumped-topological model has fewer information requirements and a much shorter execution time than a hydrodynamic model. Therefore, the approach is suitable for the simulation of long periods of time, and for comparative analysis of multiple scenarios for planning and management. Also, this type of model allows for long-term evaluation of floodplain dynamics and broader potential management implications.

The WEAP enhancements were developed for the Mompós Depression and adjacent lowland basin, with a total area of 32 198 km², or 11.8% of the total area of the entire MRB. The area receives flows from the Magdalena, Cauca, San Jorge, and Cesar rivers (Figure 1 and 5). Catchments were determined by selecting basin-scale natural "breaks" in river system topology to allow the identification of basins, inter basins, and internal basins based on the Pfaffstetter hierarchical basin coding approach (Verdin and Verdin, 1999), implemented in the recently released HydroBASINS product (Lehner and Grill, 2013). Comparison of the hydrographic units with the basin morphogenic classification (IDEAM, 2010), revealed a strong coincidence between these units. This is consistent with morphogenic classification being conditioned by factors such as geologic structure, bioclimatic conditions, topography, and slope. Land cover classification is based on seven differentiated categories in terms of their physiognomy: Forests, Shrubs, Grasslands, Agricultural Zones, Water Bodies, Hydrophytes and Others, which were derived from Colombia's ecosystem map (IDEAM et al., 2007).

### 3.2.1    Topological representation of the floodplains system

Using WEAP's semi-distributed modeling approach, Eqs. (6) to (11) can be set independently for multiple river reach and floodplain connections, allowing for the representation of complex topological relationships between catchments, river reaches, and floodplains (Figure 5). For example, a floodplain fed by the overflow from multiple river reaches, and the distribution to multiple reaches of the floodplain's return flow can both be represented.

In practice, most of the model catchment sub-units are described only by Eqs. (6) and (7), representing areas of the basin not subject to flooding. For the subset of catchments that represent floodplains (as shown in Fig. 5), the model is set up to include Eqs. (10) and (11). To reduce the number of model calibration parameters, topological connections between the river and floodplains were pre-identified using multiple sources of contextual information. In the case of the Mompós Depression, clues for permanent and episodic connectivity were derived from a review of remote sensing data (Landsat 5, 7, and 8) over time and of topological data derived from a high-resolution DEM recently developed by Colombia's [Climate] Adaptation Fund in the area between the Cauca and San Jorge rivers, as documented by Sanchez-Lozano et al. (2015).

### 3.2.2 Model calibration, validation, and uncertainty estimation

The WEAP model was calibrated (1981–1998) and validated (1999–2013) for monthly streamflow at 13 discharge gauges and for water level at four stations with long-term records in the Mompós Depression (Figure 1). Historical monthly precipitation, temperature, discharge data ($m^3 s^{-1}$), and water levels (m) were obtained from Colombia's National Meteorology, Hydrology,
and Environmental Studies Institute (IDEAM). The longest available records date back to 1940 for station QL (2903702, Calamar), located at the outlet of the Mompós Depression (Figure 1). Other stations provide relatively high serial-complete streamflow records starting in 1972.

We adopted Nash Sutcliffe Efficiency (NSE) and Relative Bias (P-BIAS) for streamflow data, and $R^2$ between wetland water levels and storage as orthogonal performance metrics; in the case of wetlands, a correlation between water levels and storage
was adopted due to a lack of topo-bathymetrical data, which prevented the conversion of the model state variable (storage volume) to effective water levels in wetland units. Despite this limitation, the $R^2$ metric reflects the model's ability to capture the dynamic character of water levels in wetland areas.

In both cases, acceptance ranges were chosen based on Moriasi et al. (2007). Model parameters (54 in total) were then calibrated using a three-stage random hypercube sampling. The first stage was derived from 10 000 simulations and the
subsequent two were derived from 1000 simulations each. Sets of model parameters above acceptance criteria ranges of 30 simultaneous metrics (13 NSE, 13 P-BIAS, and 4 $R^2$) were used to assess model uncertainty by analyzing the range of predicted average and maximum floodplain storage.

### 3.2.3 Hydrologic alteration of floodplains

One of the most widely accepted methodologies to assess the impact of changes of flow regime on aquatic ecology is the
concept of Indicators of Hydrologic Alteration (IHA), as proposed by Richter et al. (1996). IHA is a set of 32 statistics related to magnitude, timing, duration, frequency, and rate of change, which allows a detailed comparative analysis of diverse flow components. Many of the statistics are inter-correlated, rendering part of this vast amount of information redundant for high-level assessments (Gao et al., 2009; Vogel et al., 2007). In order to simplify IHA, Gao et al. (2009) demonstrated that "Ecodeficits" and "Ecosurpluses" (EDS), defined as relative changes of flow duration curves, can provide a comprehensive
simplified representation of hydrologic alteration impacts, as compared with the use of the more complex IHA approach.

In this study we employed seasonal EDS to assess the impact of variations in the hydrologic regime of wetlands storage. We divided the year into four seasons: *Subienda* (Dec–Feb), *Bajanza I* (Mar–May), *Mitaca* (Jun–Ago), and *Bajanza II* (Sep–Nov). These periods were selected based on their biologic and hydrologic relevance in the basin, in particular to fish migration, as in Jiménez-Segura et al. (2014). We differentiated ranges of duration corresponding to storage magnitude for: extreme high
(months with *percentage of time exceeded <10%: Max* to *P10*), seasonal (*P10* to *P75*), low (*P75* to *P90*), and extreme low flows (*P90* to *Min*), also relevant to diverse ecological processes (DePhilip and Moberg, 2013).

### 3.2.4 Habitat fragmentation in the upstream tributaries

We estimated fluvial length loss over the gradient 0 to 3000 masl, with a focus on reaches used by species of migratory fish present in the Mompós Wetlands. Loss of river length is a proxy for fractionation of populations and communities, and for reduction or isolation of available habitat necessary for the different life stages of species and/or groups with specific distribution ranges (Carvajal-Quintero et al., 2017; Fullerton et al., 2010).

We used biological data derived from Species Distribution Models (SDMs) fitted with MaxEnt v3.3.4 for 13 of the 16 species in the MRB known to migrate upstream from the floodplains: *Brycon henni, Brycon moorei, Curimata mivartii, Cyphocharax magdalenae, Leporinus muyscorum, Pimelodus blochii, Pimelodus grosskopfii, Plagioscion magdalenae, Prochilodus magdalenae, Pseudoplatystoma magdaleniatum, Saccodon dariensis, Salminus affinis,* and *Sorubim cuspicaudus*. Fish records consist of information available in principal ichthyological collections and surveys of migratory fish since 1940, which provide information on the historical distribution of fish in the MRB prior to hydroelectric development. A total of 31 environmental variables describing climate, soil, and geomorphology were considered. Principal Component Analysis (PCA) between those variables were used in SDMs to avoid multicollinearity. An "All Target Group" approach was used in SDMs to reduce error associated with sampling bias (Phillips et al., 2009). To evaluate model performance, we used the mean value of the area under the curve from the receptor of operator characteristic resulting from ten random cross-validation sets (70% of data for calibration and 30% for testing). The threshold that maximized the sum of specificity and sensitivity resulted from cross-validation and was used to obtain fish distributions in presence-absence format (Liu et al., 2013).

To perform connectivity analysis using the topological river network, we assigned SDMs as an attribute (presence-absence) to each river reach; as a result, a total of 11 434 km of medium and large rivers (Strahler order 4 or higher) were found to be historically associated with one or more migratory species. Migratory fish habitats are predominantly located below 1000 masl (9371 km; 85.1% of the total river network). To account for the different elevation ranges associated with different life stages of migratory fish – *Pseudoplatystoma magdaleniatum* and *Sorubim cuspicaudus* do not exceed 500 m; *Pimelodus grosskopfii* can reach 900 m; *Prochilodus magdalenae, Salminus affinis,* and *Brycon moorei* are reported to perform reproductive migrations up to elevations of 1500 m; and *Brycon henni* can reach 2000 m (Jiménez-Segura et al., 2014) – we evaluated the total loss of connectivity in three elevation ranges: 0 to 400 masl (juvenile fish growth), 400 to 1000 masl, and 1000 to 1500 masl (migration and spawning).

## 4 Results

### 4.1 Upstream impacts

#### 4.1.1 Baseline conditions

The baseline length of the river network associated with migratory fish (Strahler order ≥4) and connected to the floodplains is 6789 km in the elevation range of 0 to 400 masl, 1104 km between 400 and 1000 masl, and 123 km between 1000 and 1500

masl. Compared to a total pre-dam length of 11 434 km (6963, 2402, and 941 km, respectively, in the specified elevation ranges), this represents a loss of 28.8% of connected river length, with the greatest connectivity loss at high elevations; only 2.5% of the total river length is affected by fragmentation at 0 to 400 masl, while between 400 and 1000 masl the figure is 54.0%, and between 1000 and 1500 masl, 86.9% (Figure 6). Figure 6 illustrates the distinct differences in topographic profiles of the mainstem and its tributaries, and could be used to identify potential natural breaks in connectivity and local hotspots for endemism due to steep variations in gradient. Altitudinal distribution of fish species and habitat loss with increasing elevation is shown in Figs. 6d and 6e.

The baseline cumulative hydrologic alteration – expressed as $DOR_w$ – is 3.2%. Peñoncito station on the Magdalena River (2502733), La Coquera station on the Cauca (2624702), and La Esperanza station on the Nechí (2703701) show relatively low levels of $DOR_w$ at 5.2, 3.0, and 3.2%, respectively, but with high levels of controlled runoff: 48, 80, and 25%, respectively (Figure 7a). Current low levels of regulation are explained by the comparatively low storage capacity of existing reservoirs in comparison with basin flows. However, sediment loads are estimated to have been reduced due to reservoir trapping of 40.9, 61.3, and 39.9% at the three locations, respectively (Figure 7b).

## 4.1.2    Future scenarios

Figure 8 presents the expected cumulative impacts of 1400 generated future scenarios (1000 randomly generated – shown as grey dots, and 400 following four sets of potential basin-level restrictions- shown as colored dots), highlighting those in the range of projected expansion by 2050 (15 250±500MW). Scenarios of comparable energy capacity show wide ranges of increased cumulative impacts due to non-linearity. Regarding river fragmentation (Fig. 8a), 6763 to 4391 km of connected river length between 0 and 400 masl remain in the different scenarios (a loss of 2.9 to 36.9% from pre-dam conditions). The range of potential loss of connectivity in elevations between 400 and 1000 masl is particularly dramatic, with outcomes between 1104 and 68.5 km of remaining connected network, a 15-fold impact range. The worst-case scenario (equivalent to a loss of 97.1% with respect to pre-dam conditions) would eliminate virtually all connections between lowland floodplains and upstream spawning areas, while the best-case scenario presents no additional impacts. Figures 8b and 8c present downstream impacts of hydrologic alteration and sediment trapping, respectively. The expected range of basin-level cumulative $DOR_w$ is 4.1 to 18.1%, equivalent to 1.3 to 5.7 times the baseline condition. Ranges of additional $DOR_w$ impacts also vary substantially between the Magdalena and the Cauca, being much higher in the latter. While this is mostly a result of the relative size of Magdalena in terms of flow (the Magdalena is approximately twice the size), it is worth noting that most of the largest reservoirs projected in the basin are located in the Cauca River, which is characterized by a much narrower and steeper river valley. Cumulative sediment trapping lies in the range of 41.0 to 68.9%, representing an additional change of between 1.1 and 29% over the baseline (39.9%).

There is a wide range of expected impact associated with scenarios of comparable hydropower capacity (Figure 9). Some trade-offs in the set can be clearly identified, such as regulation between the Cauca and Magdalena (we did not attempt to

establish the pareto-optimal set, since the purpose of our study was not to perform an optimization). Through our analysis we found no statistically significant correlation between $DOR_w$ and connectivity or between $DOR_w$ and sediment trapping. This finding indicates the complementarity of the proposed metrics. In contrast, we found a high inverse correlation ($R^2$>0.84) between migratory connectivity and sediment trapping, indicating that future work could use sediment trapping as a proxy for

connectivity loss, or vice versa (Fig. 9). However, this relationship may be unique to the Magdalena system as some of the remaining basin's migratory routes are associated with large free-flowing tributaries that contribute significant sediment loads. It is worth noting that in all the scenarios considered, additional regulation in the Cauca has little to no additional effect on sediment transport reduction; this is due to the high sediment trapping of the baseline condition, and specifically due to the high sediment retention efficiency of Projects 2 and 21. (Those two projects' high sediment input will affect their longevity.)

Figures 8 and 9 also allow comparison of the range of basin-level impacts resulting from scenarios derived from the proposed sampling strategies following basin level guidelines to restrict certain projects sites. In this case, we did not attempt to explore comprehensively how different restrictions can enable better outcomes. Rather, we illustrated how a potential application of a bottom-up approach of providing key information to decision makers in the basin could enable local and individual decisions (i.e. site selection, project size, etc.) that "scale-up" to better basin-level outcomes. As shown, in some cases simple restrictions

result in expansion pathways consistently better in most of the analyzed impact and benefits metrics. In particular, as shown in Fig. 9, scenarios that avoid projects located on tributaries not yet affected by artificial barriers and mainstem projects upstream of existing reservoirs, are characterized by lower basin level impacts on all four dimensions considered. On the other hand, this type of analysis can illustrate that certain restrictions – while they may help avoid local impacts like avoiding projects affecting populated areas – are not sufficient to avoid basin-level impacts.

The five selected scenarios (highlighted in Figure 9 and summarized in Table 2) are representative of the wide range of potential boundary conditions of the Mompós Depression: A and B are equivalent in terms of low sediment trapping and fragmentation of spawning habitats, but with contrasting geographical distribution of $DOR_w$. Scenario A adds artificial regulation in the Magdalena sub-basin, B to the Cauca sub-basin. C and E correspond to "mediocre" cases, while D was in the group of worst-case scenarios in terms of impact on artificial regulation, sediment load loss, and upstream connectivity. It

should be noted that all five scenarios are plausible under Colombia's current regulatory framework.

Simulated average streamflow of the baseline and selected scenarios across stations 2502733 (Magdalena), 2624702 (Cauca), and 2703701 (Nechí) shows the two annual storage-release cycles (Storage: Mar–May and Sep–Nov, and Release: Dec–Feb and Jun–Aug), with consequent cumulative attenuation of the seasonal streamflow signal and the overall regulation effect of dry, average, and wet years (Figure 10). $DOR_w$ levels as low as 10 to 15% (corresponding to scenarios A and C for the

Magdalena and C for the Cauca), effectively reduce the amplitude of seasonal oscillations, especially in years with extreme dry macroclimatic conditions like 1992 and 1998 Niño events. Scenarios with higher $DOR_w$ (>23%) (D for the Magdalena and Cauca, and B for the Cauca), can eliminate the seasonal signal altogether in average to dry years. None of the evaluated artificial regulation scenarios affects seasonal patterns or magnitudes during wet or extremely wet periods.

It must be noted that flow alteration impacts are highly influenced by operational rules, and even reservoir configurations with a high $DOR_w$ can be operated to mimic the natural flow regime. While this study did not explore in detail the implications of alternative operational rules (our analysis only attempted to reproduce historical seasonal generation targets for the basin), the multiple simulations performed are representative of a wide range of $DOR_w$ (from 3 to 29%) and can serve as a reasonable

approximation of the envelope of expected operational behavior of multiple reservoirs with similar build-out storage capacities.

## 4.2     Floodplains analysis

### 4.2.1     WEAP model implementation

Figure 11 summarizes the model calibration and validation metrics for the sub-set (15 out of 12 000, 0.12%) of randomly generated model parameters with highest performance, above or closest to the acceptance ranges (NSE>0.65 and P-Bias<10%)

– or "good-fit" model set. As shown, performance is consistent across the 13 streamflow gauges and calibration and validation periods, with the exception of streamflow gauges 2502749 and 2502757 (calibration). However, at the same locations, performance increases during the validation period (1999–2013), which may indicate errors in the observed record at those sites during the period 1981–1998. Sharp performance decreases in gauges 2502720 and 2502764 are due to the Cauca levee breach that occurred during a 2010–2011 La Niña event.

Model sensitivity analysis of average and maximum volume storage in the main floodplain sub-units, shows results vary in the range of ±25% of the mean value of the set estimate in most of the sub-units, with the exception of C24 (Ciénaga de Ayapel), where observed variation of estimates was up to ±35% of the mean value of the subset of "good-fit" models.

*Model limitations*

The model developed runs on a monthly time step and represents large units. As a result, we were unable to evaluate high-frequency floodplain dynamics such as backwater effects on tributaries, and rates of increase in the depth and extent of flows. The extent of the flooded area was not directly reproduced by the model.

### 4.2.2     Hydrologic alteration of floodplain dynamics

Lastly, the new model allowed us to evaluate the potential changes in wetland hydrological dynamics for each of the considered

configurations of hydropower in the MRB. Figure 12 shows the simulated changes for the baseline condition and for all hydropower expansion configurations aiming for hydropower production (A to E), and alternative operation schemes aiming to reduce peak flows during extreme high events (B' and D').

Results show a heterogeneous response of the different floodplain units to upstream hydrologic alterations; units with the highest sensitivity to increased $DOR_w$ alteration are the Zapatosa, Rosario, Brazo de Loba, and Brazo Mompós, all of which

are directly influenced by the Magdalena River. The Bajo San Jorge unit, which is influenced by the San Jorge, Cauca, and Magdalena, showed a comparatively lower sensitivity to upstream hydrologic alteration. The Ayapel and San Marcos units

showed the lowest sensitivity to upstream alteration, consistent with the fact that the connection between the Cauca River and Ayapel and San Marcos floodplains became limited in the 1970s by the construction of a lateral levee west of the Cauca River (*Dique Marginal del Cauca*); currently those wetlands units are only influenced by the San Jorge River. Episodic levee failures, like the ones observed during the La Niña event of 2010–2011, have reestablished connection between the Cauca River and

the San Marcos and Ayapel systems; however, such events during extreme wet periods are not affected by dam operations, as shown in the previous section.

Low and extremely low storage events showed the highest impacts from increased regulation of upstream tributaries. Under the baseline condition and all expansion scenarios, extremely low storage events (*P90* to *min*) are expected to have much higher magnitude and be much less variable, especially in floodplains with a permanent connection between the river and

wetlands systems, like the Zapatosa, Rosario, Brazo Mompós, and Bajo San Jorge. Alteration is higher during the first half of the year, which typically oscillates with higher amplitude between dry and wet periods. Scenarios with the highest cumulative $DOR_w$ at station 2502733 on the Magdalena River (Scenario D), also induced significant changes in the magnitude of low storage events (*P75* to *P90*), modifying the amplitude of seasonal variation of floodplain and wetlands storage. Low and extremely low storage events support biodiversity by enabling several ecological processes such as reptile reproduction,

propagation of riparian vegetation communities, and nutrient and organic matter storage. Low storage also keeps invasive and introduced species in check by eliminating those that are not adapted to variable conditions.

Seasonal storage events corresponding to ranges of duration between *P10* and *P75* were found to change in floodplain units characterized by long periods of disconnection between floodplain and river systems, such as the Brazo Loba unit; with a higher sensitivity to seasonal ecodeficits in the range of *P10* to *P75* during the second half of the year, reduced seasonal storage

in this area could have severe impacts on local ecosystem functioning, as episodic yearly inundation is critical for water, nutrient, and sediment delivery to the floodplain system. Connectivity times and storage volume also determine habitat availability for migratory and resident fish.

In scenarios with the highest cumulative $DOR_w$ at station 2502733 (D), floodplain units with permanent connections like the Zapatosa, Brazo Mompós, and Rosario, also experienced small changes in storage in the range of *P10* to *P75*, and a reduction

of small seasonal flood events, potentially affecting the extent of wetlands oscillation. Seasonal oscillation also supports multiple ecosystem processes, including prevention of the invasion of riparian vegetation into the channel, and a general contribution to habitat heterogeneity.

Regarding extreme high storage events, development of hydropower dams has very low impact on high flows/flood magnitude, as extreme high flows continue to occur even under alternative operation rules focused on increased buffer capacities for

regulating extreme wet events (represented by scenarios B' and D'). None of the proposed scenarios would substantially reduce the magnitudes or duration of extreme floods associated with periodic high flow events (occurring every 10 years or more), such as those that occurred around La Niña in 2010-2011. Operation regimes aimed at maximizing energy production (maintaining higher storage to increase working head) as well as those aimed at reducing the magnitude of peak flows (maintaining lower storage to increase buffer capacity to store peak flows) show little-to-no effect on the magnitude of extreme

high events. This is consistent with the fact that even at the highest *DORw* levels (up to 39.1% in the Cauca, or 24.7% in the Magdalena), usable reservoir buffering capacity during extreme wet events can be surpassed in less than two months at peak flow volume, rendering reservoirs unable to substantially affect the magnitudes of extensive wet seasons – like those experienced during 2010 and 2011 – and forcing operators to spill water for dam safety. Nevertheless, extreme flooding events deposit nutrients and organic matter in the floodplain, recharge the water table, and determine geomorphologic dynamics of the system. As discussed in Sect. 4.1.2, proposed scenarios can also reduce sediment loads up to 69%. Through reduced sediment loads during peak flood events, wetlands and floodplains could experience reduced productivity and a progressive transformation into permanent water bodies.

## 5 Discussion

### 5.1 Contributions of this research

From a general perspective, we believe this research can contribute to the adoption of effective frameworks for strategic decision-making in the configuration of hydropower expansion. Our research shows that integrated and basin-level considerations focused on avoiding cumulative impacts on long-range, key environmental processes and components can be effectively adopted as criteria for the multiple stages of hydropower planning and development, from site selection to identification of long-term expansion potential

In this specific case, we focused our attention on key attributes of ecologically functional floodplains (Opperman et al., 2010), based on (1) hydrologic connectivity between the river and the floodplain, and between upstream and downstream sections; (2) hydrologic variability patterns and their links to local and regional processes; and (3) the spatial scale required to sustain floodplain-associated processes and benefits, like migratory fish biodiversity. Our proposed framework provides an explicit quantification of the non-linear or direct response relationship of those considerations to hydropower expansion. Changes in connectivity, hydrologic variability patterns, and the spatial scale of processes result from a wide range of scenarios that produce equivalent levels of energy generation capacity. This finding underscores the advantage of system-level integrated approaches to hydropower planning and development as well as potential to minimize impacts without sacrificing generating capacity (Hartmann et al., 2013; Nardini et al., 2016; Opperman et al., 2015), and demonstrates how consideration of the trade-offs between impacts and benefits can serve as a basis for a preventive approach. Another important finding of this research is related to how to design and evaluate transparent guiding principles that can be adopted by both policy makers and project developers; our case study illustrates some examples that take advantage of the non-linearity of impacts on freshwater systems, and explores how to inform decision makers through simple rules that can enable conditions that avoid or reduce impacts on basin-level key processes.

Another relevant contribution is the enhancements to the WEAP modeling platform to resolve water balance dynamics of floodplains and wetlands. Our study shows that the hydrologic dynamics of water storage in floodplains on a monthly to decadal scale can be represented with these enhancements. In the case of the MRB, this enables WEAP to successfully resolve

the lowland floodplains water balance at medium scales (~1000 to 10 000 km$^2$), while linking the simulation of these dynamics to upstream water management practices. By providing an improved understanding of the linkages between climate variability, system operation, and floodplain dynamics, this modelling approach can contribute in the consideration of floodplains dynamics into water management infrastructure development and operation decisions as well as in ecosystem conservation or

restoration projects.

Colombia's regulatory framework – as other countries' – currently omits any consideration of basin-level impacts of hydropower expansion; the current study provides a method to include such considerations. The wide range of scenarios, from those producing outcomes with relatively small additional environmental impacts, to those that virtually eliminate basin-level processes, provides huge potential to avoid undesirable outcomes through a comprehensive integration of system-level

performance metrics into hydropower planning. The challenge is integrating these considerations into policy design, which is currently highly reactionary and market driven.

## 5.2 Implication of the case study

The most recent analysis of sediment yield changes, performed with records from 1972 to 2010, shows no significant trend in observed sediment loads at the mouth of the Magdalena River (Restrepo et al., 2015). However, our study estimated sediment

reduction due to reservoir trapping in 1977 and 2010 at 5.3 and 18.4%, respectively, equivalent to an average decrease of 0.40% yr$^{-1}$. In addition to reservoir effects, sediment trapping must be discussed in relation to other controls on sediment yield and transport, in particular to the clearing of natural vegetation for land cultivation, which is likely to result in increased river sediment yields (Walling and Fang, 2003). Over the same period of the study of Restrepo et al. (2015), average rates of natural cover loss in the MRB were estimated at 1.4 to 1.9% yr$^{-1}$ (Etter et al., 2006; Restrepo et al., 2006). While the sediment

retention/release dynamics of the Mompós floodplains are not well understood, the apparent equilibrium in basin-level sediment transport at the river mouth might be the result of the wetlands acting to buffer the sediment balance – with sediment added from land cover change being balanced by increased retention in the wetlands and/or additional sediment trapped by reservoirs being balanced by increased sediment released from the wetlands.

Despite the uncertain contribution of the Mompós floodplains to the MRB sediment balance, we must note that the baseline

condition – which includes projects with an expected completion in 2018 – represents a significant increase in sediment trapping (from 18.4 to 39.9%) over the reference period (1972–2010) reported by Restrepo et al. (2015). Further observation of the sediment balance of the Mompós floodplain can provide more definitive evidence of project impacts. Such analysis is urgent and relevant because under certain conditions, sediment deficits could induce basin-scale system transformations, such as net subsidence of wetland and floodplain areas and a progressive transformation into permanent water bodies. The wide

range of increased sediment retention in future scenarios must also be a consideration in the assessment of hydropower contributions to carbon budgets, as studies have indicated a relationship between reservoirs' retention of organic sediments and greenhouse gas emissions (Deemer et al., 2016; Maeck et al., 2013); sediment retention is also important to the operation and longevity of hydropower dams, from a system-level perspective.

Loss of longitudinal connectivity by dams has been reported as one of the major threats to fish in the MRB, especially for migratory species and commonly fished species (Carvajal-Quintero et al., 2017; López-Casas et al., 2016). Those findings are supported by the results presented here, with the highest values of habitat fragmentation (up to 97.3%) incurred by dams situated between 400 and 1500 masl (Figure 6). Loss of longitudinal connectivity through river fragmentation could be affecting more than the migratory species evaluated here; it is important to note that this elevation range (400 to 2000 masl) contains the highest fish species richness in the MRB, including several endemic species distributed along the tributaries having the densest dam development (Carvajal-Quintero et al., 2015; Jaramillo-Villa et al., 2010). This study prioritized evaluation of the impacts of longitudinal loss, but dams and associated reservoirs also affect lateral (local) connectivity as well as vertical connectivity (connection to groundwater).

Additionally, and as illustrated, upstream hydrologic alteration can produce heterogeneous effects in the floodplain lowlands, but the most immediate consequences seem related to changes in the amplitude, magnitude, extension, and seasonal variation of floodplain inundation and wetland water storage in low- and extreme-low flow conditions (Figure 12). These, along with changes in sediment inputs due to discharge regulation in the Mompós Depression, can alter important environmental signals and stimuli for fish migration, from the floodplain to the upstream tributaries. Loss of sediment inputs – and consequently of nutrient inputs – to the floodplains, which form a nursery and feeding area for migratory fish, can affect available energy reserves for the migration and reproductive maturation essential for reproduction in the upstream tributaries, as discussed by López-Casas et al. (2016). There are other important biological effects which should be evaluated in relation to changes in the composition and functional structure of the floodplain fish assemblages in the Mompós Depression. These changes have been documented in other basins, such as the Amazon (Röpke et al., 2017). Hydrologic alteration in combination with over-fishing and habitat conversion in the lowland floodplain in the Mompós Depression could profoundly affect the food security of the people that live in the lower MRB and depend on fisheries for their food supply and income.

Our findings also reveal a distinct response of the Mompós Depression floodplains based on the relative locations of dams in the basin. Under current conditions, this system seems more sensitive to artificial regulation in the Magdalena River than in the Cauca. Hydropower in the Cauca River seems to have little additional effect in terms of alteration of floodplain inundation dynamics, as significant loss of lateral connectivity four decades ago continues to affect marshes on the west bank (the Ayapel and San Marcos). Additionally, the reservoirs of the Cauca have little influence over regulation of extreme events. This result, however, should be viewed in light of some proposals to replace the current levee on the west bank of the Cauca with infrastructure that could restore the hydraulic connection between these systems. The WEAP model developed in this study can contribute to the evaluation of such measures.

# 6    Conclusion

This paper presents a framework to quantify impacts and trade-offs to inform hydropower expansion decisions, thus enabling an integrated approach of basin-level physical, environmental, and ecosystem processes. Following Opperman et al. (2010), we focused on functional lowland floodplain systems as key basin-level environmental features, considering the impacts of

hydropower expansion on (1) hydrologic connectivity between the river and the floodplain, and between upstream and downstream sections; (2) hydrologic variability patterns and their links to local and regional processes; and (3) the spatial scale required to sustain floodplain-associated processes and benefits, like migratory fish biodiversity. Our analysis illustrates the non-linear behavior of cumulative impacts, characterized by a wide range of potential outcomes for equivalent energy expansion configurations, and demonstrates a practical approach to inform decision makers on how to design effective guidelines to effectively protect – or avoid additional impacts on – key basin-scale processes and ecosystems.

As part of this study we developed a set of enhancements to WEAP that allow for simulation of water balance dynamics of floodplains and wetlands. By providing an improved understanding of the linkages between climate variability, system operation, and floodplain dynamics, these new routines can guide the implementation of water management infrastructure development as well as ecosystem conservation or restoration projects. Both components are critical to the sustainable development of Colombia and many other countries.

From a planning perspective, we compared possible scenarios of hydropower development (as combinations of projects) that meet expected national expansion goals for 2050. In the case of the MRB, our analysis shows that baseline hydropower conditions have already significantly altered multiple basin-level processes vital to the health of the Mompós wetlands floodplains – in particular, loss of longitudinal connectivity of spawning habitats of migratory fish (-54.8%) and decreased sediment transport (-39%) – while flow regime and wetland hydrological variability maintain near natural conditions. Development scenarios, however, show a potential range up to one order of magnitude of additional impacts across comparable hydropower capacity. Some future development scenarios can result in significant physical or hydrologic alteration, i.e. a loss of longitudinal connectivity to virtually all remaining spawning habitat for migratory fish and significant reductions of sediment loads, while substantially altering floodplain (lateral) seasonal inundation dynamics in extensive areas of the Mompós Depression. Our analysis of possible scenarios, however, indicates that other scenarios would result in much lower differential changes. This emphasizes the need for comprehensive basin-level approaches to water infrastructure planning that integrate broader environmental and cumulative impacts to achieve balanced outcomes across a wide range of objectives.

We recognize that the metrics used in this analysis, while selected to provide an objective insight into multiple basin-scale key processes, are still proxies with no direct representation of the specific ecological processes of the MRB. Nevertheless, the proposed framework can serve as a basis to guide detailed studies at the reach scale to establish direct relationships.

**Acknowledgements**

This work was supported by USAID Cooperative Agreement Award No. AID-S14-A-13-00004 and AID-514-A-12-00002, the Rio Magdalena Watershed Management Program, and "Generación del Bicentenario" scholarship program funded by Colombia's *Departamento administrativo de ciencia, tecnología e innovación* - Colciencias. The authors would like to thank A. G. Hereford for her contributions in writing assistance, technical editing, language editing, and proofreading.

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

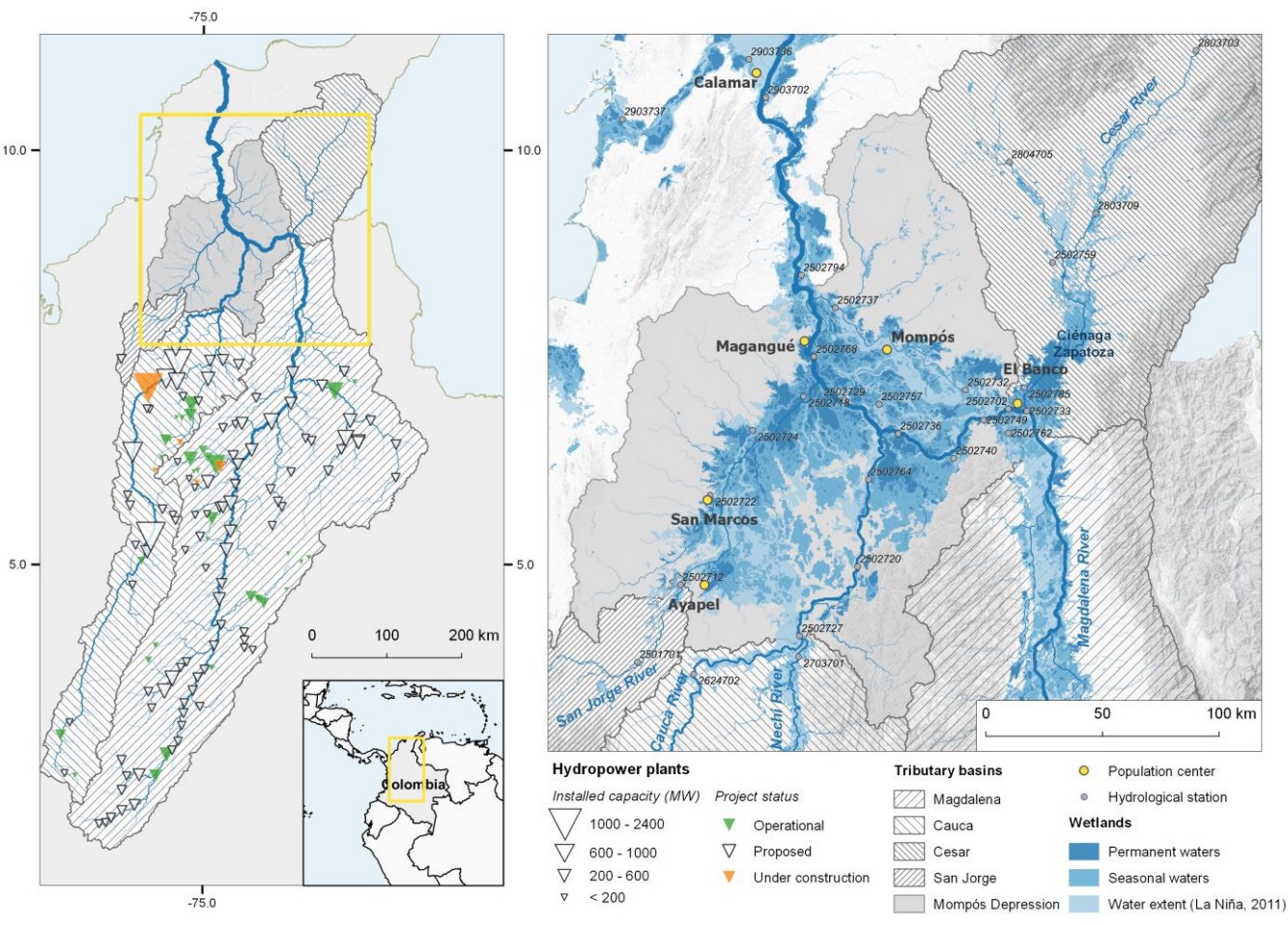

**Figure 1. Map of the Magdalena River basin showing existing and proposed hydropower dams (Left), and the Mompós Depression low floodplains system and hydrological stations (numbered) referenced in the text (Right).**

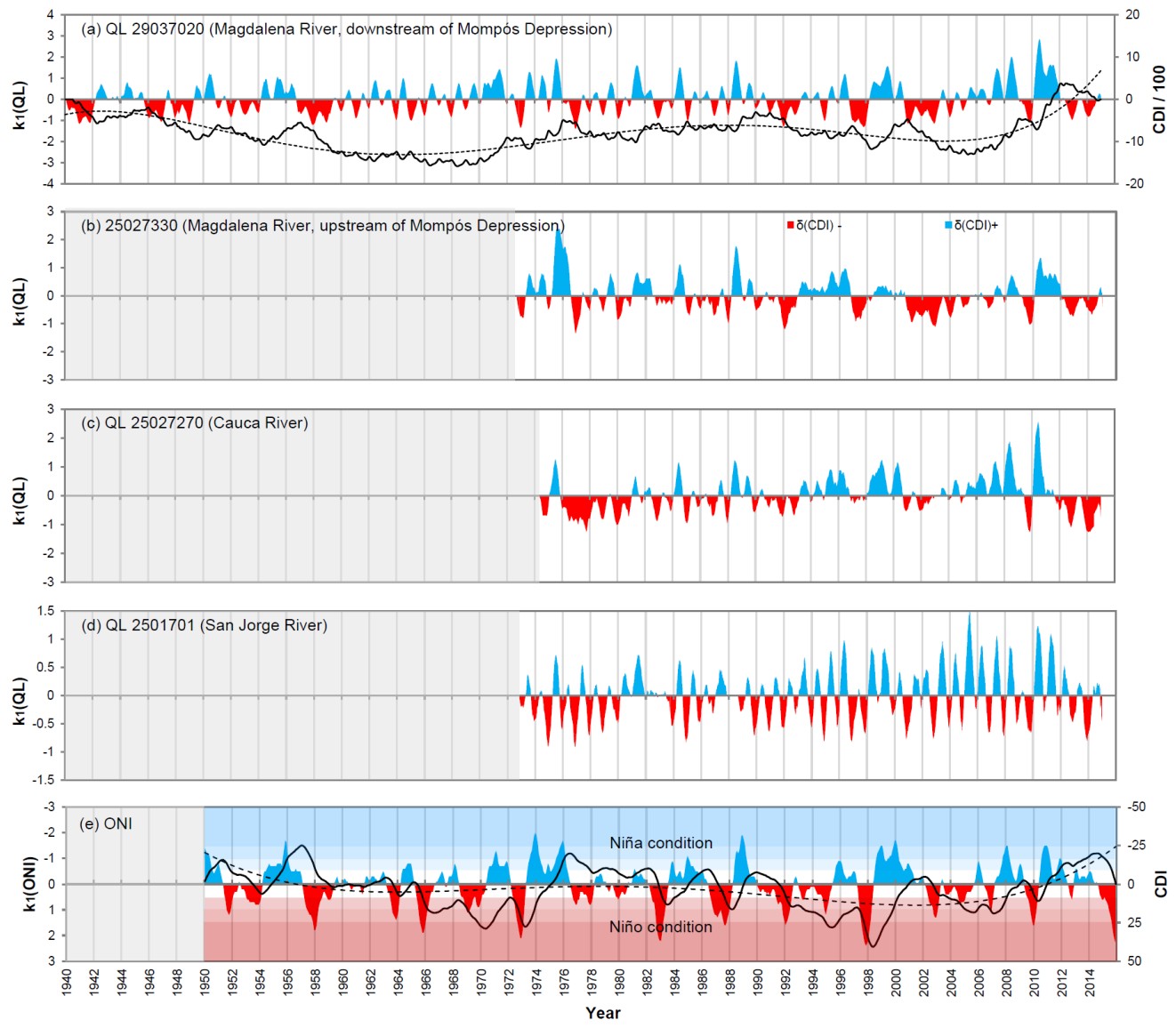

**Figure 2. Streamflow inter-annual variability, expressed as the 6-month moving average of the $k_1$ anomaly (blue and red areas) and the corresponding cumulative anomaly (continuous black line) observed at streamflow (QL) gauges (graphs a–d) in comparison to the Oceanic Niño Index (ONI; graph e) Data gaps are shaded grey.**

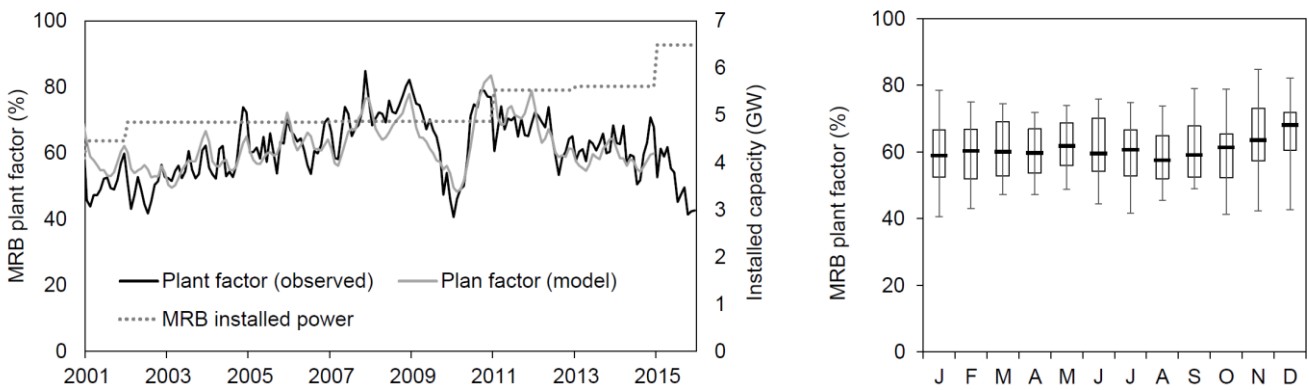

**Figure 3. Aggregated observed and modeled plant factor of Magdalena River basin (MRB) hydropower plants (2001–2015), and seasonal variation of the plant factor over the observed period.**

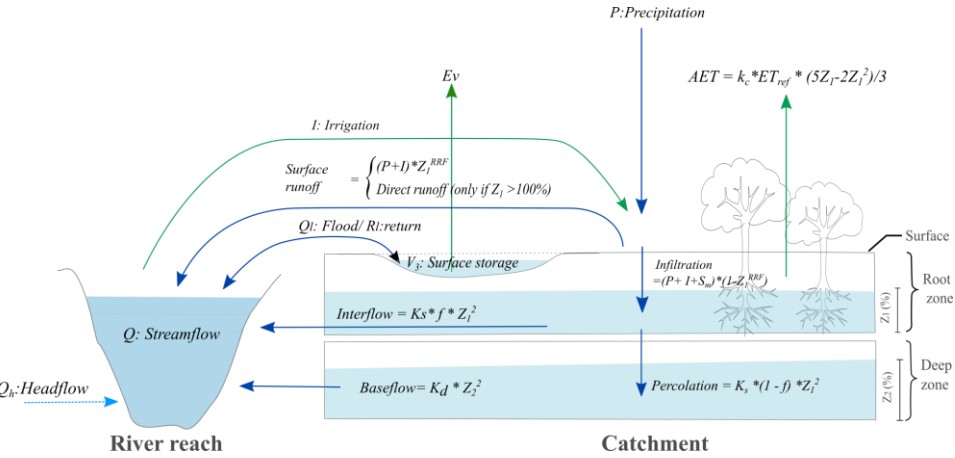

5     **Figure 4. Schematic of the enhanced two-layer soil moisture model including a surface storage component.**

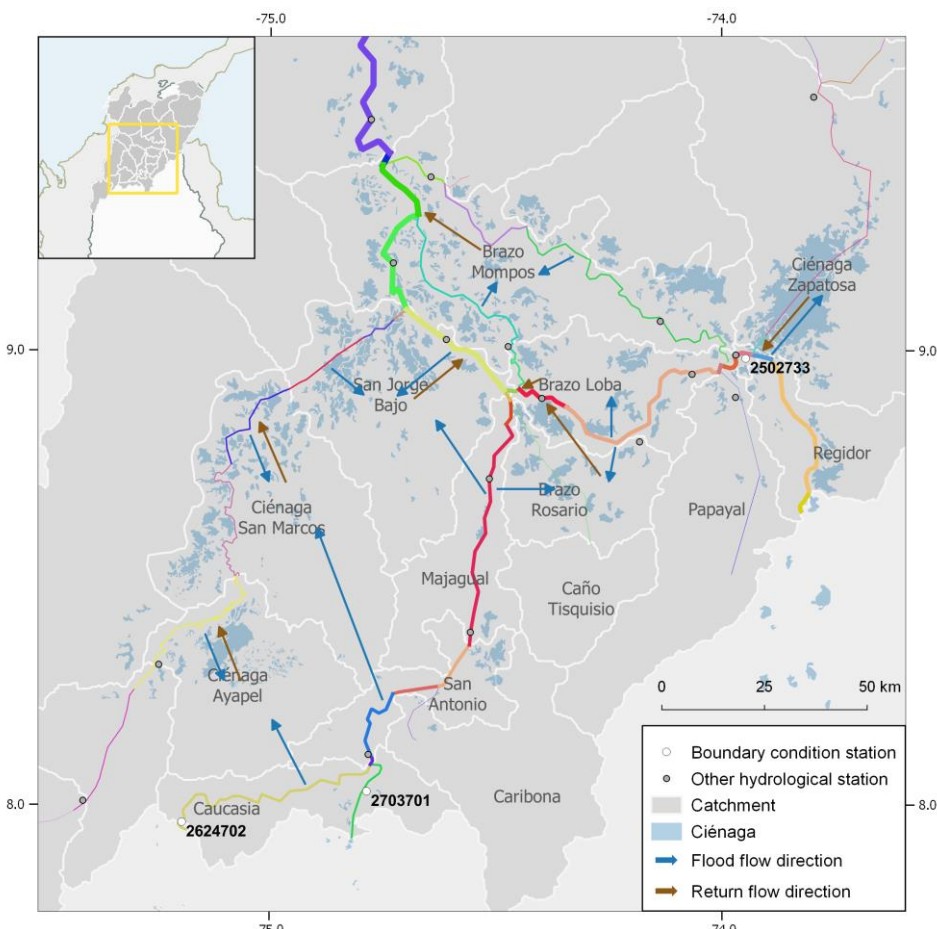

**Figure 5. WEAP Model hydrological units (catchments), river reaches (shown in different colors to illustrate the discretization of the fluvial network), and topological relationships between river reaches and wetland/floodplain areas (flood flows and returns). Stations corresponding to streamflow boundary conditions are labeled: 2502733 (Magdalena at Peñoncito), 2624702 (Cauca at La Coquera), and 2703701 (Nechí at La Esperanza).**

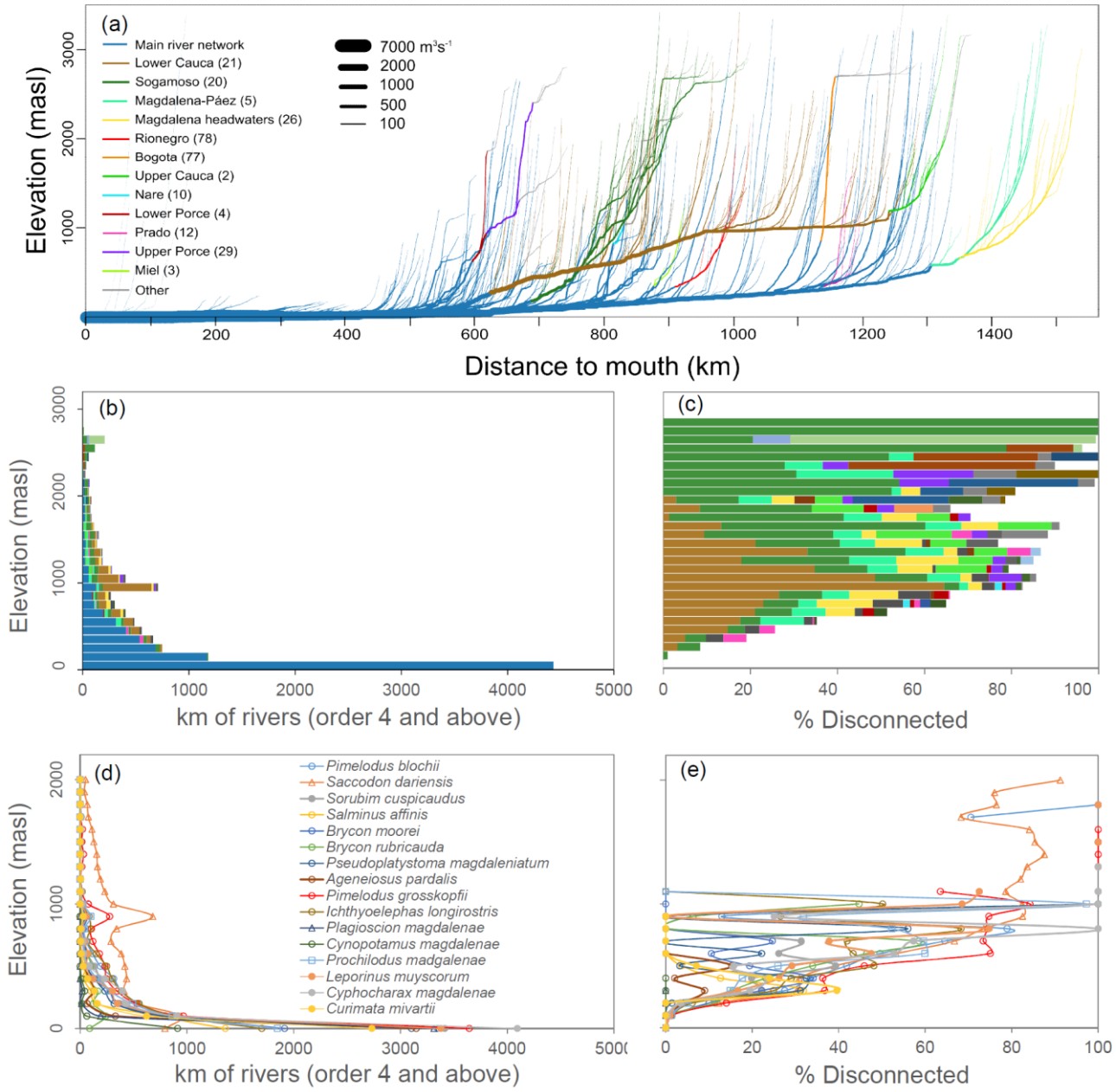

**Figure 6. Baseline conditions of remaining river network connectivity by elevation (rivers of order 4 and above). Network fragments associated with specific barriers shown in different colors (a-c; Project IDs from Table 1). Habitat availability and loss by elevation ranges of migratory fish species (d, e).**

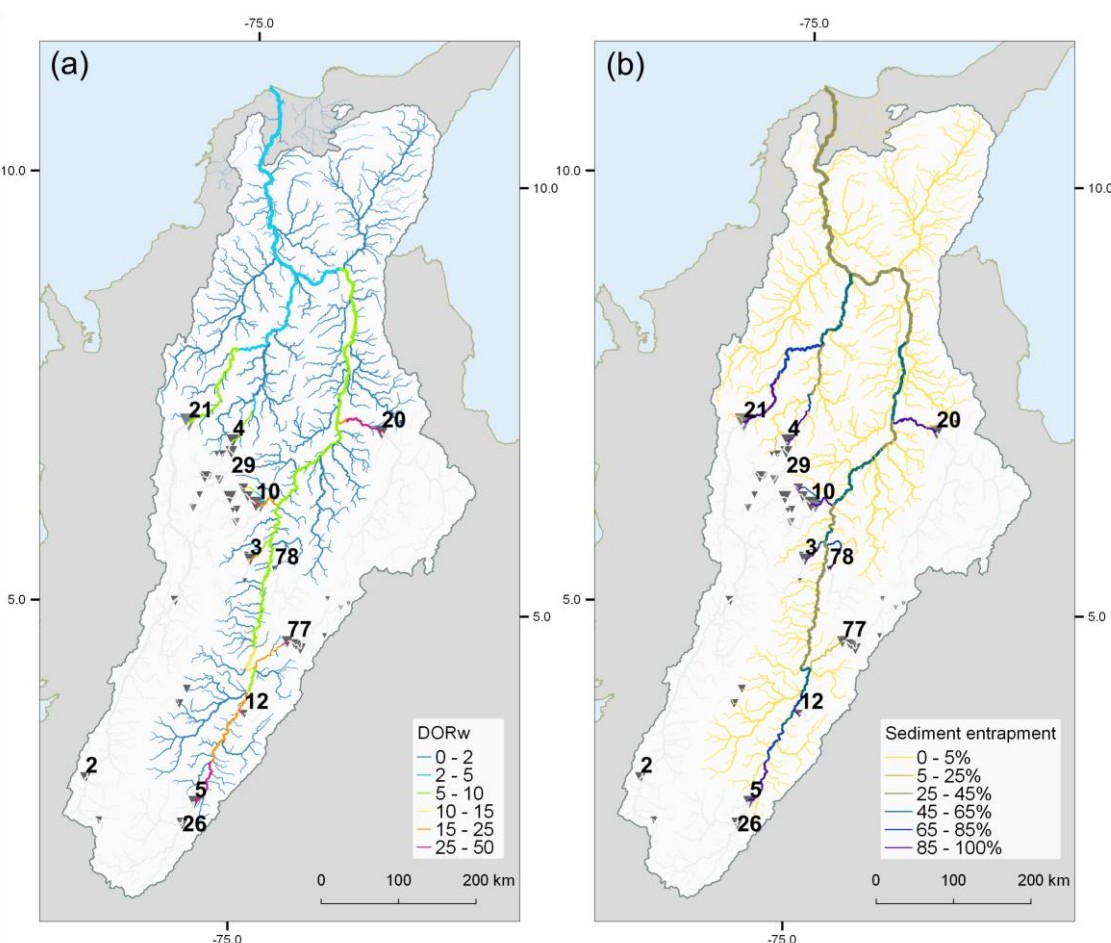

**Figure 7. Baseline cumulative impacts of existing and under-construction dams in the basin (symbolized by triangles): (Left)** *DORw* **weighted degree of regulation and (Right) percentage of sediment entrapment due to upstream reservoirs. Fragmented sections of river network are greyed out. Selected projects labeled with IDs used in Table 1.**

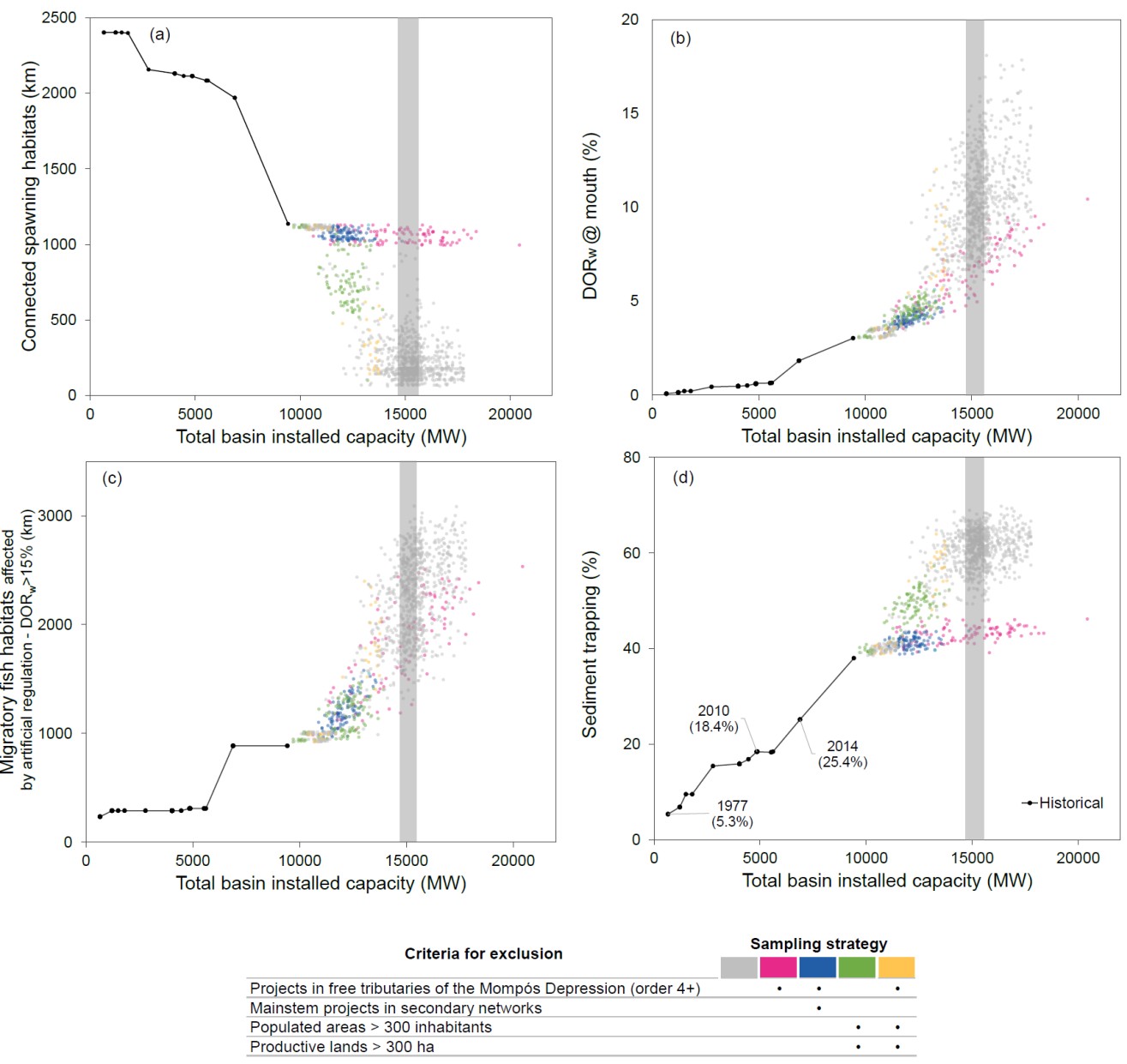

**Figure 8. Indicators of basin-level cumulative alteration of historical hydropower development and randomly generated expansion scenarios (dots) using different sampling strategies (differentiated by color). Shaded area shows the range of expected capacity by 2050 (15 250±500 MW). (a) Longitudinally connected migratory fish spawning habitat (river length) at 400–1000 masl.**
**(b) Cumulative streamflow regulation measured as *weighted degree of regulation (DOR_w)*. (c) Migratory fish habitat (river length) affected by artificial regulation (DOR>15%) (d) Total sediment trapping in reservoirs upstream of the Mompós Depression.**

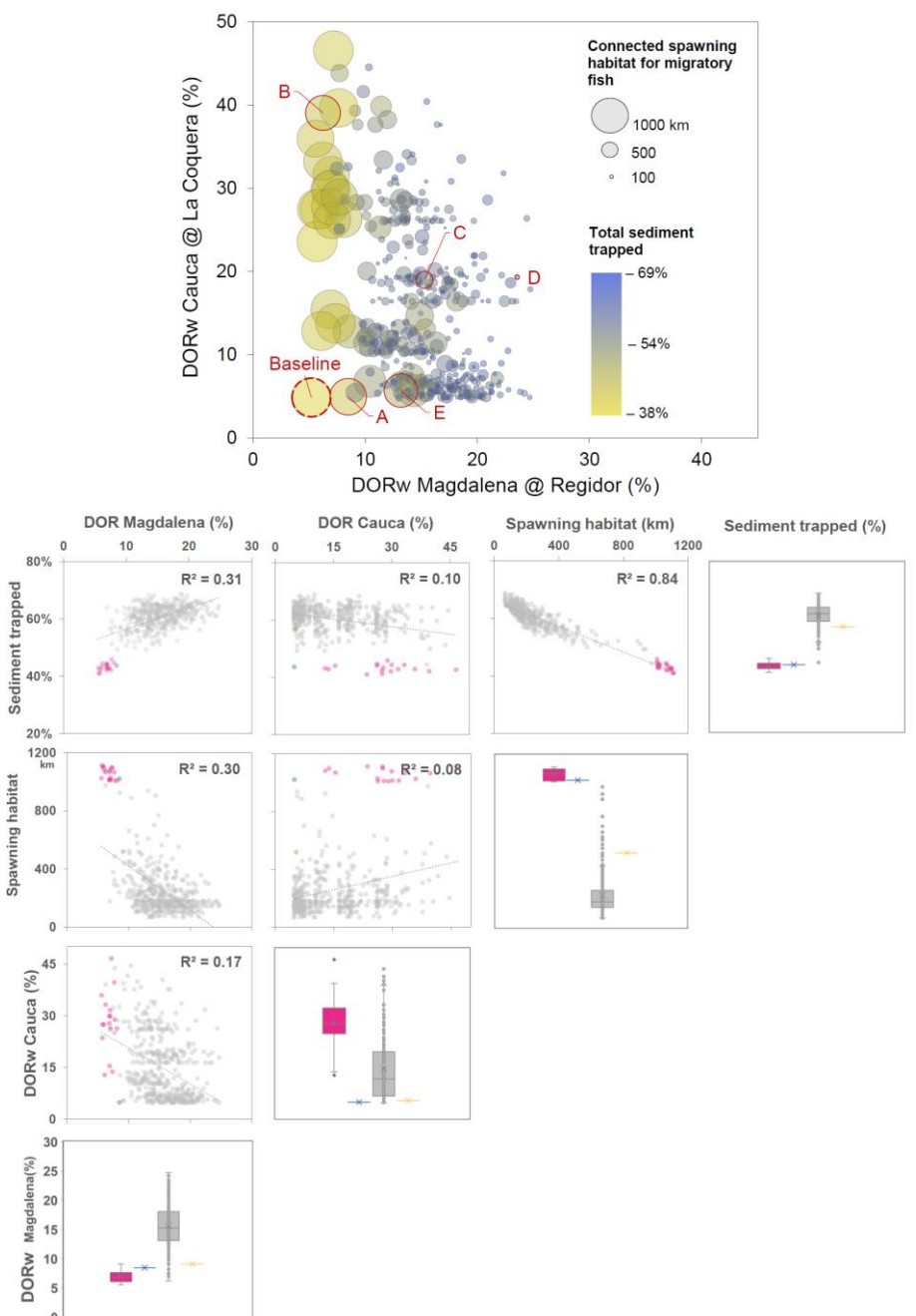

**Figure 9. Trade-off plot for scenarios in the range of expected hydropower expansion (15 250±500MW). (Top): X and Y axes are expected *DOR_w* upstream of the Mompós Depression on the Magdalena and the Cauca, respectively; bubble size represents length of connected network in the range of 400–1000 masl (spawning habitat), and color indicates the expected loss of sediment load due to reservoir trapping. Selected scenarios for detailed analyses are labeled as A, B, C, D, and E. (Bottom) 2-D plots and box-plots of individual metrics of basin-level impacts. Colors identify different sampling strategies following legend shown in Figure 8.**

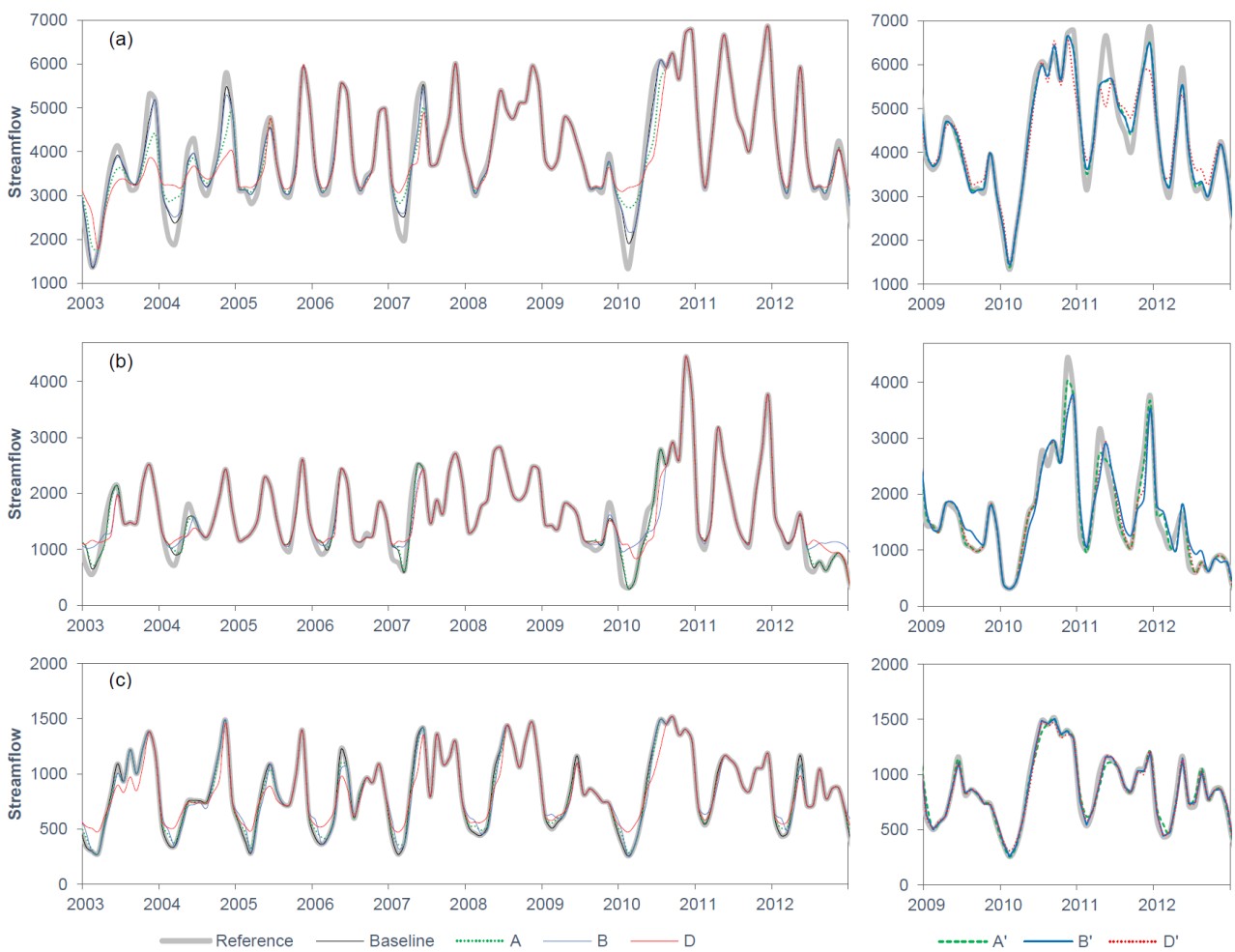

**Figure 10. Comparison of 10-year sample (2003–2012) of simulated boundary conditions (monthly average streamflow) resulting from selected hydropower configurations A, B and D. Streamflow values are shown for stations on the (a) Magdalena (2502733), (b) Cauca (2624702), and (c) Nechí (2703701) rivers, upstream of the Mompós Depression. Full period of boundary conditions is 1981– 2013. Left plots show resulting simulation from scenarios from a hydropower priority operation. Right plots show examples of operation aiming to reduce flood peak magnitudes during the Niño–Niña 2009–2011 event by maintaining low storage in reservoirs to regulate peak flows.**

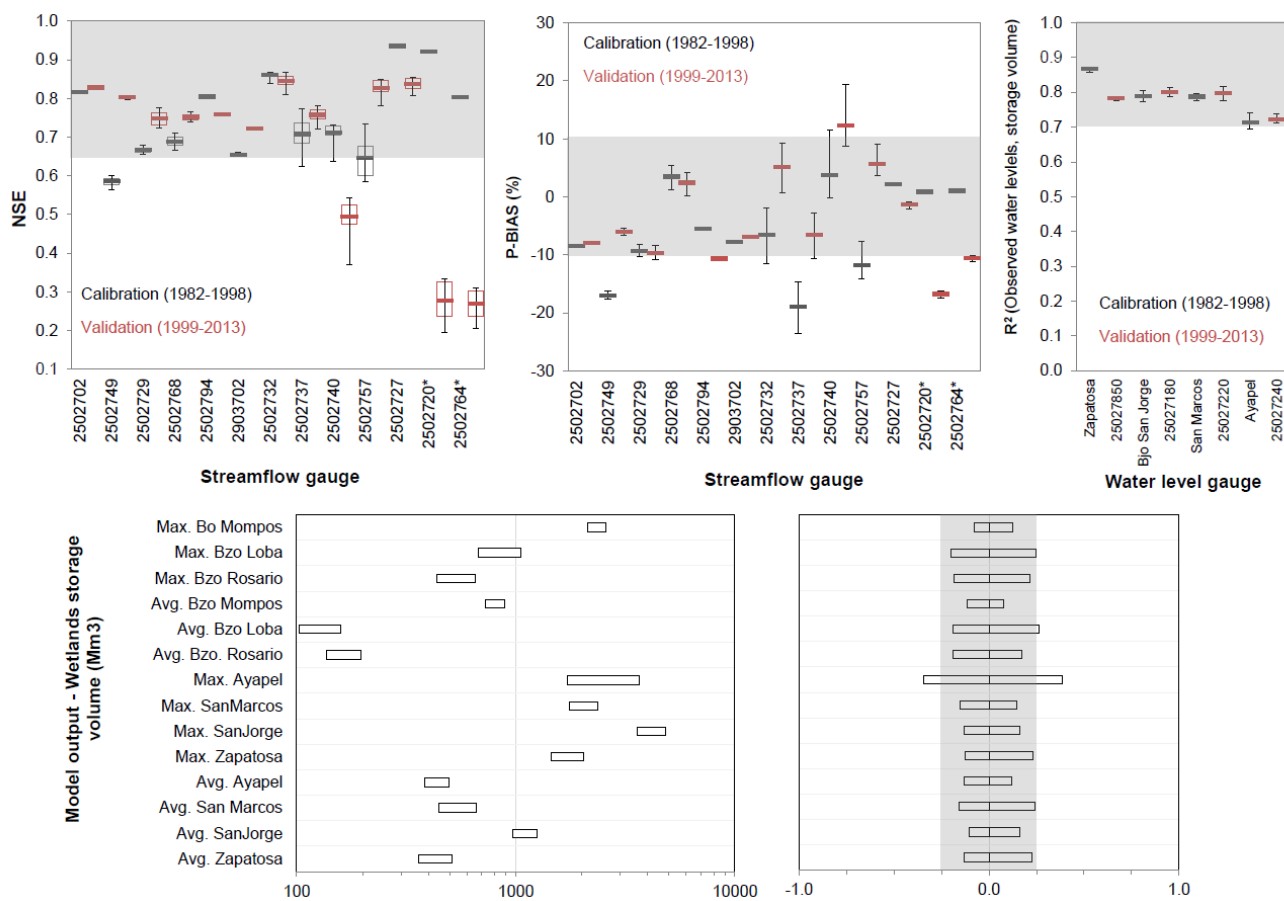

**Figure 11. Comparison of the subset of models with highest performance obtained from Monte Carlo calibration: (Above) NSE and Percentile Bias of streamflow, and Correlation Coefficient of water levels and storage volumes. Acceptance ranges highlighted in grey. (Below) Model sensitivity of "good-fit models", in terms of average and maximum volume storage in the main floodplain sub-units of the Mompós Depression.**

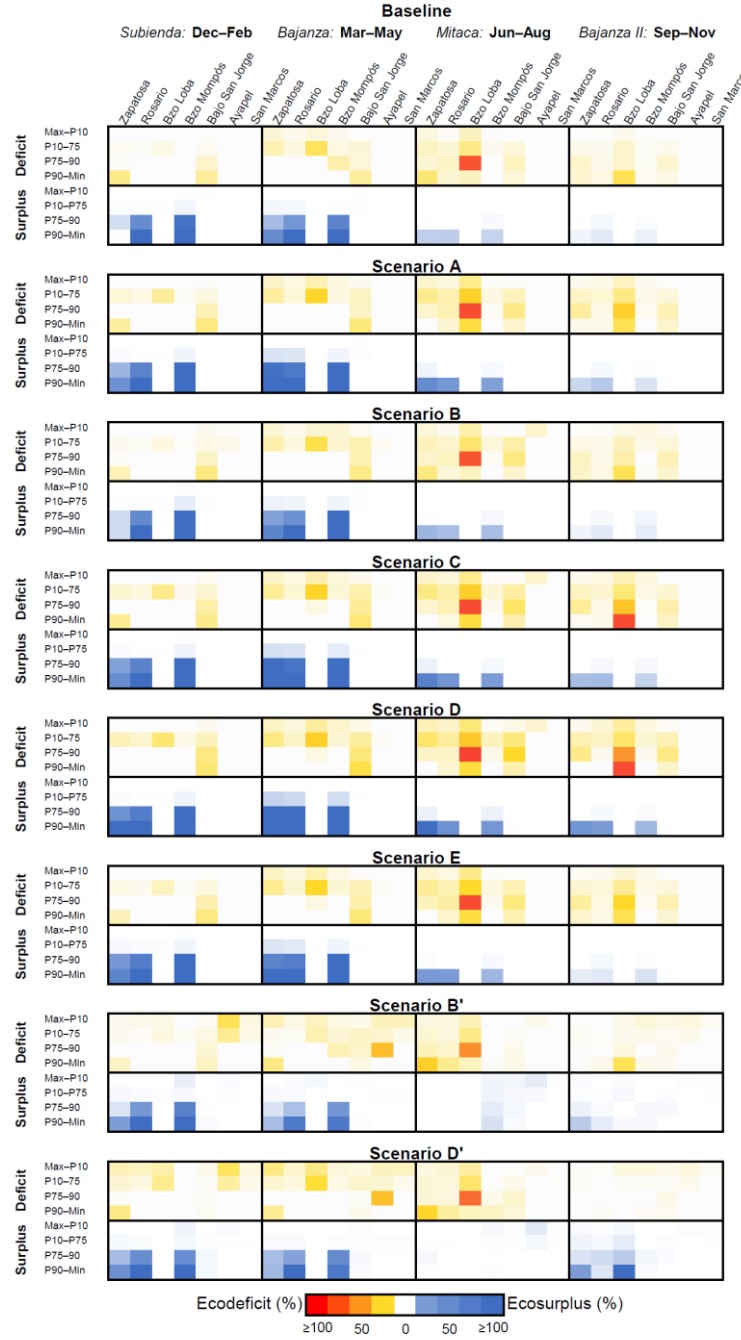

**Figure 12. Impacts of upstream regulation scenarios in wetland dynamics for the different floodplain sub-units (See locations in Figure 5), expressed as Ecodeficits or Ecosurpluses in the hydroperiod. Seasons correspond to periods of biologic and hydrologic relevance, particularly to fish migration:** *Subienda* (Dec–Feb), *Bajanza I* (Mar–May), *Mitaca* (Jun–Aug), and *Bajanza II* (Sep–Nov). **Ranges of durations representing extreme high (*Max–P10*), seasonal (*P10–P75*), low (*P75–P90*), and extreme low events (*P90–Min*) are representative of events associated with different ecological or physical processes. Scenarios B' and D' consider an alternative operational regime of configurations B and D, to reduce flood peak magnitudes by maintaining low storage in reservoirs to allow regulation of peak flows.**

**Table 1. Existing and proposed hydropower projects and other related infrastructure in the MRB, used to identify dam sets.**

| Project name | ID | Generation capacity (MW) | Gross volume estimate (million m³) | Dam height (m) | Median discharge (m³/s) |
|---|---|---|---|---|---|
| Existing | | | | | |
| Amoyá* | 42 | 80 | * | 5 | 21.2 |
| Ayurá (Transfer) | 134 | 19 | * | | 0.6 |
| Betania | 5 | 540 | 1488.0 | 58 | 388.4 |
| Cadena1_Casalaco* | 36 | 261 | * | | |
| Cadena2_Pagua* | 212 | 580 | * | | |
| Calderas | 17 | 26 | 0.0 | 25 | 7.6 |
| Canoas* | 74 | 50 | * | 0 | 117.5 |
| Carlos Lleras* | 56 | 78.2 | * | 5 | 68.3 |
| Cucuana* | 36 | 55 | * | 5 | 5.9 |
| El Colegio* | 77 | 300 | * | | 119.4 |
| Florida 2* | 121 | 24 | * | | 44.3 |
| Ituango | 21 | 2400 | 1850.0 | 197 | 1133.5 |
| Jaguas–San Lorenzo | 9 | 170 | 185.0 | 63 | 55.3 |
| Laguneta* | 76 | 80 | * | 0 | 117.7 |
| Miel | 3 | 396 | 591.0 | 188 | 118.9 |
| Miraflores | 14 | 0 | 99.0 | 0 | 4.7 |
| Muña | 1 | 270 | 0.8 | 13 | 1.2 |
| Neusa | 60 | 0 | 101.0 | 0 | 1.6 |
| Palmas | 147 | 12 | N/A | 10 | |
| Penol-Guatapé | 6 | 560 | 1071.0 | 36 | 112.2 |
| Piedras Blancas | 135 | 11 | 2.9 | 0 | 0.8 |
| Playas | 13 | 201 | 76.8 | 46 | 128.2 |
| Porce_2 | 29 | 426 | 142.7 | 118 | 172.5 |
| Porce_3 | 4 | 660 | 170.0 | 151 | 201.9 |
| Prado | 12 | 55 | 1034.0 | 92 | 113.8 |
| Quimbo | 26 | 400 | 3205.0 | 151 | 228.8 |
| Río Grande 1 | 7 | 19.9 | 0.5 | 0 | 99.5 |
| Río Grande 2 | 8 | 0 | 153.0 | 65 | 0.1 |
| Rio Negro | 78 | 10 | 13.4 | 14 | 139.3 |
| Salto I-II * | 75 | 120 | * | | 117.5 |
| Salvajina | 2 | 285 | 865.0 | 148 | 201.8 |
| San Carlos–Punchiná | 10 | 1020 | 72.0 | 70 | 145.1 |
| San Francisco | 11 | 135 | 2.3 | 8 | 0.0 |
| San Miguel | 41 | 44 | 0.3 | 5 | 102.5 |
| San Rafael (Supply storage) | 61 | 0 | 71.0 | 59.6 | 1.0 |
| Sisga | 62 | 0 | 101.2 | 0 | 2.8 |
| Sogamoso | 20 | 820 | 4800.0 | 190 | 504.0 |
| Tafetanes * | 16 | 0 | * | 0 | 2.3 |
| Tasajera * | 213 | 306 | * | 0 | 39.8 |
| Tominé (Multipurpose storage) | 63 | 0 | 690.0 | 30 | 6.9 |
| TR Guarinó (Transfer)* | 39 | 0 | * | 5 | 63.4 |
| TR Manso (Transfer)* | 40 | 0 | * | 5 | 12.0 |
| Troneras | 15 | 42 | 31.0 | 48 | 40.2 |

| Project name | ID | Generation capacity (MW) | Gross volume estimate (million m³) | Dam height (m) | Median discharge (m³/s) |
|---|---|---|---|---|---|
| Proposed | | | | | |
| Aguadas | 128 | 124 | 6.9 | 27 | 67.0 |
| Alto Saldaña | 141 | 124 | 423.2 | 155 | 97.0 |
| Ambalema | 158 | 208 | 154.4 | 19 | 1340.0 |
| Apaví | 132 | 1920 | 2639.3 | 120 | 1229.1 |
| Aranzazu | 66 | 102 | 252.6 | 120 | 119.4 |
| Atá | 142 | 109 | 197.1 | 135 | 46.0 |
| Basilio | 139 | 253 | 12680.7 | 112 | 204.2 |
| Basillas | 155 | 126 | 251.0 | 27 | 575.0 |
| Bateas | 154 | 145 | 67.4 | 31 | 520.0 |
| Bellavista | 140 | 197 | 156.6 | 57 | 109.3 |
| Boquerón | 73 | 104 | 0.6 | 22 | 30.0 |
| Buenos Aires | 69 | 106 | 1402.1 | 140 | 110.9 |
| Butantán | 79 | 268 | 1999.6 | 170 | 131.7 |
| Cabrera | 31 | 605 | 1510.1 | 177 | 327.4 |
| Cambao | 53 | 189 | 46.0 | 10 | 1260.6 |
| Cañafisto | 22 | 965 | 6487.8 | 139 | 1039.2 |
| Cañaveral | 34 | 80 | 1.0 | 32 | 19.1 |
| Carare | 117 | 582 | 1408.8 | 22 | 2287.4 |
| Carbonero | 115 | 269 | 217.4 | 14 | 2085.2 |
| Carolina | 116 | 349 | 213.1 | 16 | 2123.3 |
| Carrasposo | 156 | 150 | 151.0 | 27 | 675.0 |
| Cepitá | 103 | 172 | 19.7 | 25 | 192.0 |
| Chacipay | 86 | 164 | 310.2 | 85 | 167.1 |
| Chagualo | 137 | 100 | 188.1 | 97 | 116.5 |
| Chillurco | 149 | 161 | 359.1 | 105 | 126.0 |
| Chimurro | 120 | 146 | N/A | 0 | 27.3 |
| Cocorná | 97 | 33 | 7.0 | 42 | 22.3 |
| Coyaima | 145 | 110 | 360.8 | 34 | 246.0 |
| Cuerquia | 130 | 75 | 8.8 | 57 | 5.1 |
| El Indio | 90 | 107 | 245.6 | 70 | 125.9 |
| El Juncal | 107 | 115 | 202.1 | 27 | 421.3 |
| El Manso | 152 | 118 | 163.8 | 29 | 425.0 |
| El Neme | 143 | 480 | 5670.0 | 185 | 182.0 |
| El Palmar | 131 | 91 | 0.2 | 20 | 7.7 |
| El Tablón | 102 | 171 | 6.2 | 25 | 144.2 |
| Encimadas | 33 | 94 | 2.7 | 35 | 10.5 |
| Escuela_Minas | 38 | 55 | 0.1 | 5 | 66.2 |
| Espíritu Santo | 18 | 885 | 185.3 | 81 | 1167.5 |
| Farallones | 127 | 2120 | 11916.9 | 220 | 802.2 |
| Filo Cristal | 105 | 262 | 125.1 | 36 | 527.0 |
| Fonce | 100 | 343 | 77.7 | 65 | 113.0 |
| Furatena | 85 | 125 | 2989.9 | 115 | 122.6 |
| Guaira | 93 | 115 | 357.8 | 66 | 43.8 |
| Guane | 104 | 426 | 1063.5 | 160 | 337.5 |
| Guarapo | 148 | 104 | 533.4 | 100 | 106.0 |
| Guarquina | 94 | 69 | 60.5 | 71 | 68.8 |
| Hispania | 129 | 145 | 3.6 | 27 | 43.3 |
| Honda | 159 | 374 | 663.3 | 31 | 1370.0 |
| Horta | 88 | 114 | 1463.2 | 150 | 101.3 |
| Icononzo | 72 | 117 | 0.2 | 20 | 25.6 |
| Isnos | 64 | 103 | 33.3 | 105 | 16.3 |
| Julumito | 122 | 53 | 165.1 | 80 | 55.0 |

| Project name | ID | Generation capacity (MW) | Gross volume estimate (million m³) | Dam height (m) | Median discharge (m³/s) |
|---|---|---|---|---|---|
| La Cascada | 70 | 70 | 0.3 | 18 | 12.5 |
| La Chamba | 110 | 169 | 231.3 | 19 | 981.4 |
| La Dorada | 112 | 323 | 229.6 | 21 | 1385.3 |
| La Miel II | 35 | 120 | 0.5 | 5 | 41.1 |
| La Plata | 68 | 159 | 225.6 | 120 | 60.2 |
| La Playa | 71 | 84 | 2.9 | 25 | 22.0 |
| La Suecia | 82 | 66 | 38.0 | 100 | 14.7 |
| La Vieja | 124 | 80 | 1246.2 | 90 | 151.5 |
| Lagunilla | 83 | 60 | 0.3 | 15 | 30.9 |
| Lame | 157 | 334 | 236.6 | 28 | 1270.0 |
| Lebrija | 106 | 187 | 3269.4 | 145 | 108.4 |
| Mamaruco | 98 | 167 | 678.4 | 135 | 185.2 |
| Marañal | 113 | 461 | 612.1 | 26 | 1555.7 |
| Mayaba | 32 | 242 | 230.2 | 50 | 455.3 |
| Nariño | 50 | 356 | 118.0 | 20 | 1161.8 |
| Natagaima | 108 | 154 | 231.1 | 26 | 606.2 |
| Nus | 91 | 189 | 12.7 | 95 | 99.5 |
| Ombale | 146 | 105 | 98.0 | 34 | 238.0 |
| Oporapa | 150 | 180 | 699.7 | 130 | 130.0 |
| Páez | 67 | 143 | 81.8 | 90 | 54.2 |
| Paicol | 65 | 311 | 1570.6 | 170 | 184.2 |
| Palmalarga | 144 | 496 | 7737.3 | 160 | 296.0 |
| Palmera | 95 | 312 | 838.8 | 106 | 135.1 |
| Patagón | 114 | 170 | 102.2 | 12 | 1729.4 |
| Pericongo | 151 | 240 | 1245.6 | 120 | 136.0 |
| Piedra del Sol | 30 | 420 | 257.1 | 125 | 127.4 |
| Piedras Negras | 55 | 299 | 13.7 | 15 | 1343.1 |
| Porce 4 | 19 | 404 | 2198.1 | 195 | 223.4 |
| Porvenir 1 | 24 | 364 | 1384.9 | 167 | 166.6 |
| Porvenir 2 | 23 | 352 | 463.0 | 145 | 186.2 |
| Puente Linda | 80 | 52 | 88.9 | 90 | 55.1 |
| Riachón | 136 | 100 | 1.4 | 50 | 10.9 |
| Ricaurte | 111 | 141 | 90.8 | 16 | 1043.0 |
| Risaralda | 125 | 93 | 25.8 | 60 | 23.0 |
| Samal | 84 | 107 | 623.0 | 140 | 53.4 |
| Samaná Medio | 25 | 175 | 1668.8 | 177 | 130.3 |
| San Diego | 81 | 54 | 109.9 | 87 | 8.9 |
| San Juan | 37 | 114.3 | 0.2 | 5 | 64.6 |
| Santo Domingo | 96 | 48 | 3.1 | 23 | 35.5 |
| Simacota | 99 | 162 | 140.3 | 90 | 245.0 |
| Socotá | 101 | 124 | 1.6 | 22 | 109.2 |
| Tamar | 92 | 132 | 642.6 | 60 | 111.1 |
| Timba | 123 | 60 | 782.4 | 46 | 254.2 |
| Toloso | 133 | 334 | 167.7 | 26 | 1309.1 |
| Troya | 89 | 151 | 2341.2 | 150 | 123.8 |
| Valdivia | 138 | 700 | 728.9 | 128 | 140.6 |
| Veraguas | 153 | 110 | 202.3 | 26 | 490.0 |
| Vigía | 109 | 132 | 80.4 | 20 | 618.4 |
| Wilches | 119 | 308 | 34.8 | 11 | 3269.1 |
| Xarrapa | 126 | 330 | 351.0 | 66 | 776.7 |
| Yátaro | 87 | 150 | 1589.9 | 90 | 144.4 |
| Yondó | 118 | 308 | 129.3 | 12 | 2684.4 |

Note: Main projects are labeled with ID in Fig. 1. *: Run of river projects

**Table 2. Indices of basin-level cumulative alteration of selected scenarios at Mompós Depression boundary conditions. See Figure 5 for station locations.**

| Scenario | Installed capacity (MW) | Weighted degree of regulation (%) | | | Connected main river network (km) and loss (%) | | Cumulative sediment trapping (%) | | |
|---|---|---|---|---|---|---|---|---|---|
| | | Magdalena | Cauca | Nechí | 0–400 masl | 400–1000 masl | Magdalena | Cauca | Nechí |
| Baseline | 9781 | 5.2 | 4.8 | 0.6 | 6789 (2.5) | 1104 (54) | 40.9 | 79.2 | 24.6 |
| A | 14856 | 8.5 | 4.9 | 2.9 | 6763 (2.9) | 1021 (57.5) | 47.2 | 79.3 | 27.1 |
| B | 15603 | 6.2 | 39.1 | 4.3 | 6637 (4.7) | 975 (59.4) | 40.3 | 66.7 | 66.7 |
| C | 15081 | 15.3 | 19.0 | 2.1 | 6433 (7.6) | 485 (79.8) | 60.7 | 67.9 | 52.6 |
| D | 15635 | 23.5 | 19.3 | 24.5 | 4791 (31.2) | 143 (94.1) | 80 | 78.7 | 66.2 |
| E | 14771 | 13.2 | 5.7 | 2.9 | 6703 (3.7) | 937 (61) | 58 | 66.6 | 50.7 |