# Peer review of "Basin-scale impacts of hydropower development on the Mompós Depression wetlands, Colombia"

_Hydrology and Earth System Sciences, 2017_

## Referee Comment (RC1) · M. Arias (Referee) · 15 Oct 2017

The manuscript under discussion presents an interesting case study of impacts of hydrological alterations from potential hydropower development in the Magdalena River. The authors investigated and presented on how a large array of potential scenarios derived from random combinations of development projects plans could affect river connectivity, degree of regulation, sediment trapping, migratory fish habitat, and wetland hydrodynamics. Overall, the most significance contributions that this study provide are: (1) The configuration and analysis of the large array of modeling scenarios and (2) one of the first (to my knowledge) comprehensive scientific studies of hydropower

development in one of South America's most important rivers. The only major caveat I encountered was the vague description of the ReservoirSimulator model used, which has only been presented in conference proceedings and a thesis before. Other than that, I thought that in general the manuscript was well written, including excellent presentation of figures.

I do have, however, a number of specific comments that I suggest the authors consider in their revision:

Abstract

1. Page 1, line 14: please add the actual area (in km2) of the Mompos Depression wetland under study.

2. Page 1, line 27: episodic inundation of the floodplain during dry periods? I presume this refers to dry years rather than dry season, please clarify

Introduction

3. Page 2, line 7: first reference should be Dynesius and Nilsson, not Nilsso.

4. Page 3, line 3: when referring to the different hydrological characteristics of rivers that exhibit non-linear cumulative behavior, what does temporality refers to? Is it the same as timing?

5. Page 3, line 17: Please provide the hydropower capacity (in GW) associated with that 43% of the electricity generation in Colombia.

6. Page 3, line 29: When mentioning "large-scale" impacts (here and throughout the text), I suggest that the authors are more specific as the audience of this journal can have different interpretations of what large-scale is (continental or global?). I think "basin-scale" is the most appropriate term.

7. Page 4, line 5: Mosaic is a more appropriate ecological term than patchwork, in my opinion.

Data and Methods

8. Page 6, lines 1-4: what is the source of this information?

9. Page 6, lines 1-12: Here a general description of the ReservoirSimulator is provided, but given that such model has not been published in the international scientific peer-reviewed literature before, I wonder if this is a good opportunity to present in more depth some of the algorithms used. This could become part of the Supplementary data.

10. Page 8, line 4: What is the temporal resolution at which the Dendy's formula is used? Also, please make sure that all terms in the equation relate to the written description (capacity/inflow ratio, in particular).

11. Page 11, line 16: please make sure that the use of the terms "ecodeficit" and "ecosurplus" is correct and consistent throughout the text and figures.

Results

12. Page 13, lines 23-25: Here the authors state that they found a high inverse correlation between migratory connectivity and sediment trapping. Do you have a figure to support this? Perhaps a separate frame in Fig 9. Highlighting this finding could be important as it might be very relevant to other large tropical rivers undergoing similar tradeoffs.

13. Page 13, line 33: please add the number of scenarios (5): "It should be noted that all 5 scenarios are plausible..."

14. Page 14, line 18: here authors say the acceptance value for NSE is 0.65, but figure 11 has NSE> 0.75 shaded. Please fix.

Discussion

15. Page 16, lines 25-33: in addition to this interesting discussion on sediment trapping, could you also comment on how that could affect the operation and longevity of

hydropower dams in the Magdalena? Would there be any risks that the high rates of deforestation could make sediment accumulation much higher? Are there any measures or incentives from the hydropower sector to abate this potential issue?

16. Page 17, line 5: I believe that this is the first time the term "reference period" is mentioned, although it is presented in figure 10. A quick explanation in the methods of what the authors mean by reference versus baseline periods would be helpful.

17. Page 17, line 31: please add the word "with" to "Hydrologic alteration in combination with over-fishing. . ."

---

## Referee Comment (RC2) · Anonymous Referee #2 · 21 Dec 2017

General Comments This manuscript presents a large scale, integrated, modelling study in the Magdalena basin in Colombia, exploring the impact of different scenarios of hydropower development. The number of hydropower projects in the basin is slated to increase extensively, which will potentially lead to substantial impacts on flow regimes, reduce connectivity, and affect the extensive system of fluvial wetlands in the Mompós depression. The manuscript applies the WEAP model to evaluate the impact of different scenarios of hydropower development, based on the set of potential hydropower projects proposed, under construction, or under study in the basin. The current HP projects are also considered While I find the manuscript interesting and can see that it provides a thorough insight into the impacts of the development of hydropower poten-

tial, exemplified by the case of the Magdalena basin, I struggle to find the scientific novelty in the paper. That does not mean, however, that it is not there, but does mean that the authors should be more explicit in highlighting what scientific advances they make. To my mind the novel aspect of the paper, including the proposed enhancements to WEAP to link simulation of more local scale floodplain dynamics to basin scale scenarios, is the integrated approach that can help design basin level development strategies and make visible trade-offs that need to be made in terms of the geographic distribution of developments and impacts. To this end, I would recommend the authors elaborate how the characteristics of the scenarios influence the range of impacts on wetlands and connectivity found. What is it that alleviates impacts and what is it that exacerbates impacts? How are impacts influenced by the geographic configuration of the HP projects selected in the scenario? Do these include only small projects, or rather only large projects? What can be said about the spatial spread? Is it best to spread projects selected across sub-basins, or are one-two large tributaries "sacrificed" in lieu of low impacts in other basins? This would require a more detailed analysis of the scenarios, at least the five that have been selected from the 1000 for further study.

These questions are I think relevant as they can provide valuable insight on trade-offs in the exploitation of hydropower potential. In the work developed I think the authors do have data to provide answers to these questions at their fingertips, and taking the analysis that one step further, and developing conclusions based on that analysis could develop the paper from what now seems to be more a model application study, to a scientific contribution that warrants publication.

Overall the written English is good, but it tends to deteriorate towards the end of the manuscript. Particularly the discussion seems to have been written in a bit of a hurry. There are sufficient native English speakers in the author list to fix this.

Detailed comments: All pages: The format of the references in the main text is not in line with that of HESS, please amend.

[Figure]

Page 4 – Line 28: splits again at Calamar, with a part of the flow diverted westward to Cartagena through an altered channel system that serves as a navigation canal, a part flowing into a 100-km long delta, while the main river continues to its mouth at Barranquilla.

Page 5 – Line 4: I agree that discharge patterns are largely influenced by ICTZ. However, there are other factors that should be included in the description that influence climatic variability across the basin. This results in the bi-model character not being equally strong across the basin, with the lower basin near the Caribbean costs often suggested to be uni-modal in character. Also the role of orography is important in determining the spatial rainfall patterns, and that may be of relevance to the interpretation of the scenarios.

Page 5- Line 13: While the linkage between ENSO and climate variability in the MRB is relevant to mention, as well as other linkages such as to the PDO, I think that Figure 2 is redundant as it does not have a direct contribution to the content. Consider removing.

Page 5 – Line 30: Break the sentence as shown below to avoid the suggestion that the HP projects have as their main purpose to reduce network connectivity and produce d/s alterations! This study focused on existing and proposed medium and large hydropower projects, including reservoirs and run-of-river plants. These can reduce river network connectivity or produce downstream alteration.

Page 6 – Line 8: It would be useful to elaborate a little on the construction of the scenarios. What as the strategy for sampling from the possible 97 projects? I assume that the selection of some projects would preclude the selection of other projects, or are they all fully independent? How were the five scenarios selected? Was this at random or was some strategy employed? This should be elaborated.

Page 7 – Line 26: I assume that the moving average operator is applied over a six month period. It may be good to add this clarification

Page 9 – Line 13: I have been looking at equations 6-9 and cannot quite figure these out. What are these equations based on? Are these physical water balances, or are these empirical relationhips? It seems to me that the equations are also not consistent in dimensions, particularly with respect to time. There is a capital Z and a small z – are these the same?

Page 10 – Lines 5-10: The authors present the algorithms they have used to improve the ability of WEAP to model surface flows. I agree that a conceptual approach would appear adequate in this case, particularly given the monthly time step. However, this more conceptual model does require extensive parameterisation. A good example is the thresholds that are mentioned in equations 10 and 11. How have these been derived? Are these based on topological information, or is this a parameter that derived through calibration? If so, then how was that calibration carried out, and were comparisons against more complete hydrodynamic models done (given that these are available in the basin)?

Page 11 – Line 4: I am not sure I understand what the authors mean when saying that R2 is determined between water levels and storage. Should this not be between simulated and observed water levels?

Page 12 – Line 13: I am somewhat confused by the river lengths, which are reported for pre- and post dam conditions. It is noted here that the length of the river network with Strahler > 4 is 10373. However, on the same page in the results 8311 km post-dam and 11998 km pre-dam length is reported for Strahler > 4. I guess the number on this line is the total river length to the mouth, while the second is the total length of river, unobstructed by a dam from the limit of the Mompos floodplains. A clearer indication of what length is being discussed would help.

Page 13 – Line 6: change sentence to "sediment loads are estimated to have been reduced due to reservoir trapping of . . .."

Page 13 – Line 20: I would not say the points are random that are shown in figure 9.

As I understand it from the figure caption these are the points that comply with the HP expansion target (the same as in the shaded area in Figure 8. Correct?

Page 13 – Line 21: Drop "however" in the parenthesis.

Page 13 – Lines 19-34: What may also be interesting to mention is that the range of DORw for the Cauca is much larger than for the main Magdalena, for those scenarios that comply with the HP expansion objectives. This is likely due to the simple fact that the Madgalena is the larger of the two in terms of flow. It may also be due to the specific selection of projects. So while these may be equivalent to some extent, the difference in impact is quite distinct. What surprises me is that in Figure 10 the difference between Scenario B and for example Scenario A are not that distinct, despite the significant difference in degree of regulation (admittedly this may be due it being difficult to discern the different lines in the figure). There also seems to be very little change in all scenarios on high flow conditions, despite high degree of regulation (which implies a significant amount of storage).

Page 14 – Line 17 – 26: As noted earlier, were any comparisons made with the extent of inundation in the Mompós depression; either from observed flood extent (such as for the 2010-2011 event), or for model simulations using e.g. more complex HD models that do exist for the basin (which may be easier to relate to the more conceptual nature of the model presented here). The use or R2 for comparing water levels reveals very good statistics; but I would argue that correlation of monthly levels in a highly seasonal river would be expected to yield good similarity in simulated and observed patterns. Is there any information on the bias of the simulated levels at these four points?

Page 15 – Lines 3- 33: In the discussion on the impacts on the floodplains during high flow events it is noted that these are small, which is also reflected in the minor change to high flows. It is mentioned that there is little control over flood events due to dam safety. While I agree that this may be the case in the current situation in the basin, with a relatively low DORw, as that increases, which essentially means that the

storage volume increases, that may well change significantly. To study the possible effect the operational rules would need to be amended, as likely these have not been implemented in WEAP to consider flood control.

Page 16 – Lines 12 onwards: To my mind the discussion should also include reflections on the research that has been presented, the interpretation of the results and the contribution to the state of science. Also, the authors should reflect on some of the limitations of the approach presented (such as for example the assumptions made on the reservoir operating rules). Currently the discussion discusses primarily the relevance of the work in the context of the developments, and addresses topics beyond the scope of research. This comes back to my general comments, where I think the authors should try to upscale their finding to what these may mean to the wider (scientific) community. What are the new insights that the approach they propose provide. I do think that there are quite some that could be highlighted. To my mind the integrated approach offers handles to make strategic choices on the configuration of HP development at the basin scale. Based on an that improved discussion, the conclusions could be revised as appropriate.

Page 20-24: The authors refer extensively to "grey" literature sources such as project reports etc. Please make sure that relevant details are included. An example is ESEE (1979), where only the title is included. There are several more. Please revise.

Figure 12: The yellow colours are difficult to read. I also struggle to understand what he small white marks indicate, if anything.

---

## Author Comment (AC1) · 26 Jan 2018

We would like to thank the reviewers for their time and for their thoughtful comments. The comments led to some additional analysis and substantial improvements to the manuscript. Please find below our responses to each comment.

Note: updated version of the manuscript is included with the responses.

0. The only major caveat I encountered was the vague description of the Reservoir-Simulator model used, which has only been presented in conference proceedings and a thesis before. Author response: A description of the Reservoir simulator model has

been included as supplementary material. Changes in Manuscript: Added section SI-1

1. Page 1, line 14: please add the actual area (in km2) of the Mompos Depression wetland under study. Author response: Fixed Changes in Manuscript: Now reads: "one of the largest wetland systems in South America at 3400 km2"

2. Page 1, line 27: episodic inundation of the floodplain during dry periods? I presume this refers to dry years rather than dry season, please clarify Author response: We meant the seasonal oscillation of wetlands during dry years. Changes in Manuscript: the manuscript now reads: "similar magnitude to existing fluxes involved in the episodic inundation of the floodplain during dry years and"

3. Page 2, line 7: first reference should be Dynesius and Nilsson, not Nilsso. Author response: Fixed. Changes in Manuscript: Reference now reads Dynesius and Nilsson.

4. Page 3, line 3: when referring to the different hydrological characteristics of rivers that exhibit non-linear cumulative behavior, what does temporality refers to? Is it the same as timing? Author response: Yes, "timing" is what was meant. Changes in Manuscript: Now reads "changes in the magnitude, frequency, duration, and timing of river..."

5. Page 3, line 17: Please provide the hydropower capacity (in GW) associated with that 43Author response: Agreed, we adjusted the text to include that information. Changes in Manuscript: Now reads: "Medium and large hydropower plants in the Magdalena River basin (MRB) with a total capacity of 6.89 GW currently supply 49

6. Page 3, line 29: When mentioning "large-scale" impacts (here and throughout the text), I suggest that the authors are more specific as the audience of this journal can have different interpretations of what large-scale is (continental or global?). I think "basin-scale" is the most appropriate term. Author response: Agreed Changes in Manuscript: All instances of "large-scale" changed to "basin-scale"

7. Page 4, line 5: Mosaic is a more appropriate ecological term than patchwork, in

my opinion. Author response: Agreed. Changes in Manuscript: Manuscript now reads "The Magdalena River is located in the Northern Andes Mountains and drains a biodiverse mosaic of ecosystems. . ."

8. Page 6, lines 1-4: what is the source of this information? Author response: Data is provided by XM, company in charge of the operation of the Colombian energy market. Changes in Manuscript: The corresponding reference was included.

9. Page 6, lines 1-12: Here a general description of the ReservoirSimulator is provided, but given that such model has not been published in the international scientific peer-reviewed literature before, I wonder if this is a good opportunity to present in more depth some of the algorithms used. This could become part of the Supplementary data. Author response: Agree. Changes in Manuscript: A section "Supplementary Information" has been added with a description of the ReservoirSimulator routines

10. Page 8, line 4: What is the temporal resolution at which the Dendy's formula is used? Also, please make sure that all terms in the equation relate to the written description (capacity/inflow ratio, in particular). Author response: Dendy's formula is based on annual data. Capacity is C and Inflow I. Changes in Manuscript: Now reads "Dendy's method is a revised Brune curve, which uses an empirical expression to estimate the long-term average reservoir sediment retention efficiency based on the ratio between capacity (C) and average annual inflow (I). A higher ratio indicates higher sediment retention efficiency, TE, as described by the following equation: TE=100*(0.97)^{(0.19^{Log(C/I)}}

11. Page 11, line 16: please make sure that the use of the terms "ecodeficit" and "ecosurplus" is correct and consistent throughout the text and figures. Author response: Thank you for this note. Changes in Manuscript: We went through the document to verify consistency of the terms and made changes as necessary.

12. Page 13, lines 23-25: Here the authors state that they found a high inverse correlation between migratory connectivity and sediment trapping. Do you have a figure

to support this? Perhaps a separate frame in Fig 9. Highlighting this finding could be important as it might be very relevant to other large tropical rivers undergoing similar tradeoffs. Author response: We updated Figure 9, to include a more detailed display of trade-offs between the considered impacts. Changes in Manuscript: Figure 9 changed.

13. Page 13, line 33: please add the number of scenarios (5): "It should be noted that all 5 scenarios are plausible..." Author response: Change made. Changes in Manuscript: Manuscript now reads "It should be noted that all five scenarios are plausible..."

14. Page 14, line 18: here authors say the acceptance value for NSE is 0.65, but figure11 has NSE> 0.75 shaded. Please fix. Author response: Change made. Changes in Manuscript: Corrected figure.

15. Page 16, lines 25-33: in addition to this interesting discussion on sediment trapping, could you also comment on how that could affect the operation and longevity of hydropower dams in the Magdalena? Would there be any risks that the high rates of deforestation could make sediment accumulation much higher? Are there any measures or incentives from the hydropower sector to abate this potential issue? Author response: We do not have the full data needed to assess the rate of capacity loss of dams due to sediment trapping. In particular, we don't have data on sediment bedload transport. However, we added mention of the need to consider sediment trapping in the planning of site locations due to comparative rates of sediment inflow (see noted change below). As for the impact of deforestation, it would likely increase sediment accumulation, but such an analysis is beyond the scope of this project. Changes in Manuscript: We appended the results section with the following: "It is worth noting that in all the scenarios considered additional regulation in the Cauca has little to no additional effect on sediment transport reduction; this is due to the high sediment trapping of the baseline condition, and specifically due to the high sediment retention efficiency of Projects 2 and 21. As a result, those projects will be subject to high sediment input that will affect their longevity."

16. Page 17, line 5: I believe that this is the first time the term "reference period" is mentioned, although it is presented in figure 10. A quick explanation in the methods of what the authors mean by reference versus baseline periods would be helpful. Author response: Agreed. Changes in Manuscript: We indicated the reference period in section 3.1.1. The text now reads: "From the subset of scenarios that meet projected hydropower expansion by year 2050—an equivalent hydropower capacity of 15.25±0.5 GW, or +125

17. Page 17, line 31: please add the word "with" to "Hydrologic alteration in combination with over-fishing..." Author response: Change made. Changes in Manuscript: Manuscript now reads "Hydrologic alteration in combination with over-fishing and habitat conversion..."

Please also note the supplement to this comment:
https://www.hydrol-earth-syst-sci-discuss.net/hess-2017-544/hess-2017-544-AC1-supplement.pdf

—————————————

[Figure]

**Supplement:**

[revised manuscript text omitted]

**SI-1. SUPPLEMENTARY INFORMATION**

**ReservoirSimulator model description**

The ReservoirSimulator allows simulation of the water balance of a group of reservoirs, based on individual or system-level dispatch rules for three main categories of use: downstream instream requirements, hydropower, and other supply (supply that bypasses turbines).

For a given reservoir, the model takes into account physical and technical constraints, such as the volume-elevation curve, tail-water elevation, operational levels (inactive, buffer, technical, and safety), turbine type, capacity, and efficiency. It allows modeling of the topology of reservoirs and tributary sub-basins.

ReservoirSimulator performs a lumped, discrete-time water balance of the components shown in Figure SI-1 with the model parameters and inputs shown in Table SI-1.

For each reservoir, calculations are done according to the sequence in Table SI- 2:

[Figure]

**Figure SI- 1: Schematic of reservoir inflow, outflow and storage components of the reservoir simulator model.**

**Table SI-1: Description of reservoir data requirements of the model**

| Symbol | Reservoir physical and technical data | Units |
|---|---|---|
| $Q_a(t)$ | Specific runoff: Runoff from upstream sub-watershed during step $t$ (excludes outflows from upstream infrastructure) | $m^3/s$ |
| $\Delta t$ | Time step | s |
| $V_c$ | Reservoir storage capacity at top of conservation level ($Z_c$) | $m^3$ |
| $V_{buffer}$ | Reservoir storage capacity at buffer level ($Z_b$) | $m^3$ |
| $V_{min}$ | Reservoir storage capacity at top of inactive level ($Z_d$) | $m^3$ |
| $V_{w0}$ | Reservoir storage capacity at floodgates operative level ($Z_w$) | |
| $H(V)$ | Volume-elevation curve of reservoir | masl |
| $A(V)$ | Area-elevation curve of reservoir | masl |
| $B_c$ | Buffer coefficient: fraction of buffer storage that can be allocated in a given time step *Note: Can be a user defined function of local (i.e. reservoir storage, etc.) or system-level state variables or context variables (i.e. downstream storage, etc.).* | % |
| $B_w$ | Floodgates allocation factor: determines the fraction of storage above the floodgates operational level delivered downstream in a given time step. *Note: Can be a user defined function of local (i.e. reservoir storage, etc) or system-level state variables or context variables (i.e. downstream storage, etc.).* | |
| $ET(t)$ | Evaporation rate in a time step $t$ | mm |
| $Z_0$ | Tail water elevation (water level at the point where turbine flow is discharged) | masl |
| $D(t)$ | Water demand diverted from the reservoir, bypassing the turbines, with higher priority than hydropower | $m^3$ |
| $V_{eco,1}(t)$ | Average instream flow requirement at step $t$, for the reach between reservoir and turbines discharge | $m^3$ |
| $P$ | Installed generation capacity | Mw |
| $f$ | Friction loss factor in hydropower load pipes. | m |
| $N$ | Number of turbines | |
| $E(H,Q)$ | Turbine efficiency curve, as a function of net head and flow (typically a function of turbine type i.e. Francis, Pelton, etc.) | % |
| $e_{target}$ | Minimum efficiency threshold of turbines. Hydropower won't be generated if actual efficiency is lower than $e_{target}$ | % |
| $Q_w$ | Floodgates max hydraulic capacity | $m^3/s$ |
| Reservoir topological data | | |
| $K_T$ | Identifier of the project downstream of turbine flow discharge | |
| $K_E$ | Identifier of the project downstream of e-flow discharge | |
| $K_s$ | Identifier of the project downstream of spills discharge | |
| $R$ | Position of the reservoir in the simulation sequence (1st, 2nd, …) | |

**Table SI- 2: Sequence of calculations performed by the reservoir simulator model at a given time-step**

| Value | Description | Calculation | Units |
|---|---|---|---|
| $Q_u(t)$ | Sum of upstream outflows (turbine, e-flows and/or spills) directed to the reservoir | | $[m^3/s]$ |
| $S_t$ | Reservoir storage at the beginning of step t | | $[m^3]$ |
| $A_t$ | Area of reservoir at the beginning of step t, calculated from area volume curve | $A(S_t)$ | $[m^2]$ |
| L | Reservoir losses as evaporation | $A_t * ET$ | $[m^3]$ |
| $Va_t$ | Total available water for allocation without restrictions at the reservoir, during step t | $\max\big(0, S_t + (Q_a(t) + Q_u(t)) * \Delta t - L(t) - V_{buffer}\big)$ | $[m^3]$ |
| $Vr_t$ | Total available water for allocation, from the buffer zone, during step t | $(V_{buffer} - V_{min}) * B_c$: if $Va_t > 0$
$(S_t + Q_a(t) * \Delta t - L(t) - V_{min}) * B_c$: if $Va_t = 0$ | |
| $Vu_t$ | Total available water for allocation during step t | $Vr_t + Va_t$ | |
| $Z_t$ | Reservoir water level at the beginning of step t, calculated from volume elevation curve | $Z(S_t)$ | masl |
| $\Delta Z_t$ | Working water head at step t. | $Z_t - Z_0$ | $[m]$ |
| $H_f$ | Friction losses | $\Delta Z_t * f$ | $[m]$ |
| RPE | Expected generation at step t, expressed as a percentage of installed capacity. | User defined function of local (i.e. reservoir storage at previous timestep, instream flow requirement downstream of turbines, etc) or system-level state variables or context variables (i.e. system level demand, ENSO signal, etc.). See paper Figure 3 | $[\%]$ |
| $V_{pt}$ | Release requirement, to fulfill 100% of expected energy generation at step t, operating at target efficiency | $RPE * P * 1e6 / (9801 * (\Delta Z_t - H_f) * e_{target}) * \Delta t$ | $[m^3]$ |
| $Vd_t$ | Total water available during step t, after the allocation of demands with higher priorities than hydropower | $\max(0, Vu_t - D_t - V_{eco,1})$ | $[m^3]$ |
| $V_{te}$ | Effective turbined volume during step t | $\min(V_{pt}, Vd_t t)$ | $[m^3]$ |
| $Q_{te}$ | Effective turbined flow during step t | $V_{te}/\Delta t$ | $[m^3/s]$ |
| $Vs_t$ | Water available after hydropower is allocated. | $Vd_t - Vt_e$ | $[m^3]$ |
| $V_w$ | Controlled spill volume (floodgates operation) | $V_w = \min\big(R_w^{1/B_w}, Q_w * \Delta t\big)$
with:
$R_w = \max(0, Vs_t - V_{w0})$: Volume above the floodgates operational level | $[m^3]$ |
| $Vs_w$ | Water available after controlled spill is allocated. | $Vs_t - V_w$ | $[m^3]$ |
| $V_{spill_t}$ | Total spill during step t | $\max(0, Vs_w - Vc)$ | $[m^3]$ |
| $P_n$ | Net power output if available flow is turbined given $H_{n,t}$, using n turbines. | $\gamma * \dfrac{Q_{te}}{n} * H_{n,t} * e_{t,n}$ | $[Mw]$ |
| $e_{t,n}$ | Actual Turbine/Generator efficiency given $(\Delta Z_t - H_f)$ and flow $= \frac{Q_{te}}{n}$. See example in | $e_n$
$0$ if $e_n < e_{target}$ | % |

| Value | Description | Calculation | Units |
|---|---|---|---|
| | Figure SI-2. If efficiency is lower than a target efficiency, is adopted as 0. | | |
| $P_t$ | Actual power output for step t | $\max(P_1, 2 * P_2, \dots N * P_N)$ | [Mw] |
| $S_{t+1}$ | Reservoir storage at the beginning of step t | $Vs_t - V_{spill_t}$ | $m^3$ |

[Figure]

**Figure SI-2 Example of a turbine power-efficiency function (Francis type)**

---

## Author Comment (AC2) · 26 Jan 2018

Note: Updated manuscript and SI-1 (Supplementary Information) are attached

Below are reviewers' comments, each immediately followed by the Author Response and notes about related Changes in the Manuscript:

General comments: This manuscript presents a large scale, integrated, modelling study in the Magdalena basin in Colombia, exploring the impact of different scenarios of hydropower development. The number of hydropower projects in the basin is slated to increase extensively, which will potentially lead to substantial impacts on flow

regimes, reduce connectivity, and affect the extensive system of fluvial wetlands in the Mompós depression. The manuscript applies the WEAP model to evaluate the impact of different scenarios of hydropower development, based on the set of potential hydropower projects proposed, under construction, or under study in the basin. The current HP projects are also considered While I find the manuscript interesting and can see that it provides a thorough insight into the impacts of the development of hydropower potential, exemplified by the case of the Magdalena basin, I struggle to find the scientific novelty in the paper. That does not mean, however, that it is not there, but does mean that the authors should be more explicit in highlighting what scientific advances they make. To my mind the novel aspect of the paper, including the proposed enhancements to WEAP to link simulation of more local scale floodplain dynamics to basin scale scenarios, is the integrated approach that can help design basin level development strategies and make visible trade-offs that need to be made in terms of the geographic distribution of developments and impacts. To this end, I would recommend the authors elaborate how the characteristics of the scenarios influence the range of impacts on wetlands and connectivity found. What is it that alleviates impacts and what is it that exacerbates impacts? How are impacts influenced by the geographic configuration of the HP projects selected in the scenario? Do these include only small projects, or rather only large projects? What can be said about the spatial spread? Is it best to spread projects selected across sub-basins, or are one-two large tributaries "sacrificed" in lieu of low impacts in other basins? This would require a more detailed analysis of the scenarios, at least the five that have been selected from the 1000 for further study. These questions are I think relevant as they can provide valuable insight on trade-offs in the exploitation of hydropower potential. In the work developed I think the authors do have data to provide answers to these questions at their fingertips, and taking the analysis that one step further, and developing conclusions based on that analysis could develop the paper from what now seems to be more a model application study, to a scientific contribution that warrants publication.

Author response: We would like to thank the reviewer for these thoughtful comments.

[Figure]

The comments led to some additional analysis and substantial improvements to the manuscript. In summary the main changes are: - We performed a more detailed analysis on sampling approaches to define expansion scenarios, considering both basin-level or project-level characteristics. Results are discussed in terms of the implications for policy design. - We performed a more detailed analysis of the five selected scenarios in terms of the implications of operational approaches on system hydrologic dynamics. - Discussion and conclusions were appended to highlight other scientific advances and general implications of the presented research.

All pages: The format of the references in the main text is not in line with that of HESS, please amend. Author response: HESS style added to Mendeley and selected for use. Changes in Manuscript: All citations now in HESS style.

Page 4 – Line 28: splits again at Calamar, with a part of the flow diverted westward to Cartagena through an altered channel system that serves as a navigation canal, a part flowing into a 100-km long delta, while the main river continues to its mouth at Barranquilla. Author response: Change made. Changes in Manuscript: Manuscript now reads ". . .until the Magdalena splits again at Calamar, with part of the flow diverted westward to Cartagena through an altered channel system that serves as a navigation canal and part flowing into a 100-km long delta, while the main river continues to its mouth in Barranquilla."

Page 5 – Line 4: I agree that discharge patterns are largely influenced by ICTZ. However, there are other factors that should be included in the description that influence climatic variability across the basin. This results in the bi-model character not being equally strong across the basin, with the lower basin near the Caribbean costs often suggested to be uni-modal in character. Also the role of orography is important in determining the spatial rainfall patterns, and that may be of relevance to the interpretation of the scenarios. Author response: Agreed. Paragraph was revised to mention other factors controlling the climate variability patterns in the basin. Changes in Manuscript: Added text: "The roles of topography, soil-atmosphere interactions, the Atlantic Ocean,

and the Amazon also influence temporal and spatial rainfall patterns, resulting in the bimodal character not being equally strong across the basin (Poveda et al., 2011). The lower basin near the Caribbean coast—including the Mompós depression—is often suggested to be unimodal in character, and the southeastern portion of the basin (approximately below 2°N) is characterized by a distinct unimodal pattern, with a June-to-August wet season.

Page 5- Line 13: While the linkage between ENSO and climate variability in the MRB is relevant to mention, as well as other linkages such as to the PDO, I think that Figure 2 is redundant as it does not have a direct contribution to the content. Consider removing. Author response: Figure 2 illustrates the hydroclimatic variability components present in the major tributaries of the Mompós Depression. These components are mentioned several times throughout the manuscript, making this figure useful for readers who are not familiar with the Magdalena River Basin. Changes in Manuscript: None.

Page 5 – Line 30: Break the sentence as shown below to avoid the suggestion that the HP projects have as their main purpose to reduce network connectivity and produce d/s alterations! This study focused on existing and proposed medium and large hydropower projects, including reservoirs and run-of-river plants. These can reduce river network connectivity or produce downstream alteration. Author response: Change made. Changes in Manuscript: Manuscript now reads "This study focused on existing and proposed medium and large hydropower projects, including reservoirs and run-of-river plants. Such projects can reduce river network connectivity or produce downstream alteration."

Page 6 – Line 8: It would be useful to elaborate a little on the construction of the scenarios. What was the strategy for sampling from the possible 97 projects? I assume that the selection of some projects would preclude the selection of other projects, or are they all fully independent? How were the five scenarios selected? Was this at random or was some strategy employed? This should be elaborated. Author response: In the new version of the manuscript, we incorporated additional elements in the identification

of the scenarios that allow for a more rich interpretation and discussion regarding the guiding principles that determine better basin-scale outcomes. Changes in Manuscript: Section 3.1.1. was fully rewritten.

Page 7 – Line 26: I assume that the moving average operator is applied over a six month period. It may be good to add this clarification Author response: Indeed. Changes in Manuscript: Now reads: "...and MA6 a moving average operator applied over a six-month period"

Page 9 – Line 13: I have been looking at equations 6-9 and cannot quite figure these out. What are these equations based on? Are these physical water balances, or are these empirical relationhips? It seems to me that the equations are also not consistent in dimensions, particularly with respect to time. There is a capital Z and a small z – are these the same? Author response: These are empirical functions. Steps were added to clarify the dimensions. Z and z are the same. Changes in Manuscript: Clarification on basis of equations: "The model "uses empirical functions that describe evapotranspiration, surface runoff, sub-surface runoff or interflow, and deep percolation" (Yates et al., 2005a, p.491)." See new equations 8 and 9. Z's have been corrected to be z's.

Page 10 – Lines 5-10: The authors present the algorithms they have used to improve the ability of WEAP to model surface flows. I agree that a conceptual approach would appear adequate in this case, particularly given the monthly time step. However, this more conceptual model does require extensive parameterisation. A good example is the thresholds that are mentioned in equations 10 and 11. How have these been derived? Are these based on topological information, or is this a parameter that derived through calibration? If so, then how was that calibration carried out, and were comparisons against more complete hydrodynamic models done (given that these are available in the basin)? Author response: Thanks for the comment. There are two steps in setting up the model—first, to identify which areas of the model will be represented by floodplains and what are the topological connections. This is done using contextual information and does not require calibration. This first step significantly reduces the

parameters to be calibrated. The second step consists of doing a monte-carlo calibration as described in section 3.2.2. No comparisons were made with other models as there aren't any accessible hydrodynamic model results that fully cover our study area. Changes in Manuscript: A second paragraph was added to section 3.2.1, Topological representation of the floodplains system, to describe the procedure used to identify the topology of the river and floodplains.

Page 11 – Line 4: I am not sure I understand what the authors mean when saying that R2 is determined between water levels and storage. Should this not be between simulated and observed water levels? Author response: There is not enough topo-bathymetric information available to make a direct comparison of modeled and observed water levels in the wetland. We still consider R2 a useful metric to characterize the skill of the model to represent the dynamic characteristics of water storage. Model outputs of water storage dynamics are very sensitive to parameterization, so R2, in addition to NSE and P-Bias at streamflow gauges, contributes to identifying more competent models. We recognize that this is an approximation. In Section 4.2.1, we discuss the implications of this in terms of the sensitivity of the model outputs. Changes in Manuscript: "in the case of wetlands, a correlation between water levels and storage was adopted due to a lack of topo-bathymetrical data, which prevented the conversion of the model state variable (storage volume) to effective water levels in wetland units. Despite this limitation, the R2 metric reflects the model's ability to capture the dynamic character of water levels in wetland areas."

Page 12 – Line 13: I am somewhat confused by the river lengths, which are reported for pre- and post dam conditions. It is noted here that the length of the river network with Strahler > 4 is 10373. However, on the same page in the results 8311 km post-dam and 11998 km pre-dam length is reported for Strahler > 4. I guess the number on this line is the total river length to the mouth, while the second is the total length of river, unobstructed by a dam from the limit of the Mompos floodplains. A clearer indication of what length is being discussed would help. Author response: Correct—previous

version mixed total connected river length and length used by migratory fish. Changes in Manuscript: Numbers were changed so that all reflect connectivity of migratory fish habitat, rather than total length. Additionally, the numbers have been updated to reflect the inclusion of 3 additional migratory fish species.

Page 13 – Line 6: change sentence to "sediment loads are estimated to have been reduced due to reservoir trapping of ...." Author response: Change made. Changes in Manuscript: Manuscript now reads: "However, sediment loads are estimated to have been reduced due to reservoir trapping of. . ."

Page 13 – Line 20: I would not say the points are random that are shown in figure 9. As I understand it from the figure caption these are the points that comply with the HP expansion target (the same as in the shaded area in Figure 8. Correct? Author response: Correct. Changes in Manuscript: We clarified in the Methods Section that the points are a defined subset of the randomly generated scenarios. "Each set is a randomly sampled subset of projects (colored dots in Figure 8) that avoid one or more criteria—projects located on Mompós Depression tributaries (order 4+) not yet affected by artificial barriers, mainstem projects. . ."

Page 13 – Line 21: Drop "however" in the parenthesis. Author response: Change made. Changes in Manuscript: Manuscript now reads "(we did not attempt to establish. . ."

Page 13 – Lines 19-34: What may also be interesting to mention is that the range of DORw for the Cauca is much larger than for the main Magdalena, for those scenarios that comply with the HP expansion objectives. This is likely due to the simple fact that the Madgalena is the larger of the two in terms of flow. It may also be due to the specific selection of projects. So while these may be equivalent to some extent, the difference in impact is quite distinct. What surprises me is that in Figure 10 the difference between Scenario B and for example Scenario A are not that distinct, despite the significant difference in degree of regulation (admittedly this may be due it

being difficult to discern the different lines in the figure). There also seems to be very little change in all scenarios on high flow conditions, despite high degree of regulation (which implies a significant amount of storage). Author response: We simplified the plots to facilitate the comparison of the relevant scenarios. Updated plots show differences between scenarios A, B and D. Small changes in high flow conditions are the result of the adopted operational regime focused on maintaining high water levels to increase head and turbine efficiency. We appended the analysis to include two alternative operational scenarios focused on regulation of extreme high flows, by reducing the operational level of reservoirs and allowing a higher buffer volume for regulating peak flows. Changes in Manuscript: Updated figures 10 and 12, and results section.

Page 14 – Line 17 – 26: As noted earlier, were any comparisons made with the extent of inundation in the Mompós depression; either from observed flood extent (such as for the 2010-2011 event), or for model simulations using e.g. more complex HD models that do exist for the basin (which may be easier to relate to the more conceptual nature of the model presented here). The use or R2 for comparing water levels reveals very good statistics; but I would argue that correlation of monthly levels in a highly seasonal river would be expected to yield good similarity in simulated and observed patterns. Is there any information on the bias of the simulated levels at these four points? Author response: There is not full topo-bathymetric data for the entire study area, so it is not possible to perform such analysis. However, the R2 metric was found to be very sensitive to the model parameterization, and in most of the calibration trials values were found not satisfactory. Changes in Manuscript: None.

Page 15 – Lines 3- 33: In the discussion on the impacts on the floodplains during high flow events it is noted that these are small, which is also reflected in the minor change to high flows. It is mentioned that there is little control over flood events due to dam safety. While I agree that this may be the case in the current situation in the basin, with a relatively low DORw, as that increases, which essentially means that the storage volume increases, that may well change significantly. To study the possible

effect the operational rules would need to be amended, as likely these have not been implemented in WEAP to consider flood control. Author response: We performed additional analysis to explore impacts of alternative operational rules focused on reducing peak flow events. We included three new sub-scenarios for the previously identified configurations of dams that followed these alternative operational rules. Changes in Manuscript: These additional sub-scenarios (A', B', and D') are now included in Figures 10 and 12, and in the discussion.

Page 16 – Lines 12 onwards: To my mind the discussion should also include reflections on the research that has been presented, the interpretation of the results and the contribution to the state of science. Also, the authors should reflect on some of the limitations of the approach presented (such as for example the assumptions made on the reservoir operating rules). Currently the discussion discusses primarily the relevance of the work in the context of the developments, and addresses topics beyond the scope of research. This comes back to my general comments, where I think the authors should try to upscale their finding to what these may mean to the wider (scientific) community. What are the new insights that the approach they propose provide. I do think that there are quite some that could be highlighted. To my mind the integrated approach offers handles to make strategic choices on the configuration of HP development at the basin scale. Based on an that improved discussion, the conclusions could be revised as appropriate. Author response: Thanks for the comment. We reorganized the discussion section to highlight in more depth the contribution to the state of science and the broader implications of the integrated approach. Changes in Manuscript: Added Section: 5.1. Contributions of this research into the discussion chapter. Modified the conclusions section.

Page 20-24: The authors refer extensively to "grey" literature sources such as project reports etc. Please make sure that relevant details are included. An example is ESEE (1979), where only the title is included. There are several more. Please revise. Author response: Agreed Changes in Manuscript: We updated the reference style to include
[Figure]

relevant details (such as publishing institution and weblink where available).

Figure 12: The yellow colours are difficult to read. I also struggle to understand what the small white marks indicate, if anything. Author response: We increased the contrast in the pallete, and eliminated the white marks Changes in Manuscript: Updated figure.

Please also note the supplement to this comment:
https://www.hydrol-earth-syst-sci-discuss.net/hess-2017-544/hess-2017-544-AC2-supplement.pdf
* * *
[Figure]

**Supplement:**

[revised manuscript text omitted]

**SI-1. SUPPLEMENTARY INFORMATION**

**ReservoirSimulator model description**

The ReservoirSimulator allows simulation of the water balance of a group of reservoirs, based on individual or system-level dispatch rules for three main categories of use: downstream instream requirements, hydropower, and other supply (supply that bypasses turbines).

For a given reservoir, the model takes into account physical and technical constraints, such as the volume-elevation curve, tail-water elevation, operational levels (inactive, buffer, technical, and safety), turbine type, capacity, and efficiency. It allows modeling of the topology of reservoirs and tributary sub-basins.

ReservoirSimulator performs a lumped, discrete-time water balance of the components shown in Figure SI-1 with the model parameters and inputs shown in Table SI-1.

For each reservoir, calculations are done according to the sequence in Table SI- 2:

[Figure]

**Figure SI- 1: Schematic of reservoir inflow, outflow and storage components of the reservoir simulator model.**

**Table SI-1: Description of reservoir data requirements of the model**

| Symbol | Reservoir physical and technical data | Units |
|---|---|---|
| $Q_a(t)$ | Specific runoff: Runoff from upstream sub-watershed during step $t$ (excludes outflows from upstream infrastructure) | $m^3/s$ |
| $\Delta t$ | Time step | s |
| $V_c$ | Reservoir storage capacity at top of conservation level ($Z_c$) | $m^3$ |
| $V_{buffer}$ | Reservoir storage capacity at buffer level ($Z_b$) | $m^3$ |
| $V_{min}$ | Reservoir storage capacity at top of inactive level ($Z_d$) | $m^3$ |
| $V_{w0}$ | Reservoir storage capacity at floodgates operative level ($Z_w$) | |
| $H(V)$ | Volume-elevation curve of reservoir | masl |
| $A(V)$ | Area-elevation curve of reservoir | masl |
| $B_c$ | Buffer coefficient: fraction of buffer storage that can be allocated in a given time step *Note: Can be a user defined function of local (i.e. reservoir storage, etc.) or system-level state variables or context variables (i.e. downstream storage, etc.).* | % |
| $B_w$ | Floodgates allocation factor: determines the fraction of storage above the floodgates operational level delivered downstream in a given time step. *Note: Can be a user defined function of local (i.e. reservoir storage, etc) or system-level state variables or context variables (i.e. downstream storage, etc.).* | |
| $ET(t)$ | Evaporation rate in a time step $t$ | mm |
| $Z_0$ | Tail water elevation (water level at the point where turbine flow is discharged) | masl |
| $D(t)$ | Water demand diverted from the reservoir, bypassing the turbines, with higher priority than hydropower | $m^3$ |
| $V_{eco,1}(t)$ | Average instream flow requirement at step $t$, for the reach between reservoir and turbines discharge | $m^3$ |
| $P$ | Installed generation capacity | Mw |
| $f$ | Friction loss factor in hydropower load pipes. | m |
| $N$ | Number of turbines | |
| $E(H,Q)$ | Turbine efficiency curve, as a function of net head and flow (typically a function of turbine type i.e. Francis, Pelton, etc.) | % |
| $e_{target}$ | Minimum efficiency threshold of turbines. Hydropower won't be generated if actual efficiency is lower than $e_{target}$ | % |
| $Q_w$ | Floodgates max hydraulic capacity | $m^3/s$ |
| Reservoir topological data | | |
| $K_T$ | Identifier of the project downstream of turbine flow discharge | |
| $K_E$ | Identifier of the project downstream of e-flow discharge | |
| $K_s$ | Identifier of the project downstream of spills discharge | |
| $R$ | Position of the reservoir in the simulation sequence (1st, 2nd, …) | |

**Table SI- 2: Sequence of calculations performed by the reservoir simulator model at a given time-step**

| Value | Description | Calculation | Units |
|---|---|---|---|
| $Q_u(t)$ | Sum of upstream outflows (turbine, e-flows and/or spills) directed to the reservoir | | $[m^3/s]$ |
| $S_t$ | Reservoir storage at the beginning of step t | | $[m^3]$ |
| $A_t$ | Area of reservoir at the beginning of step t, calculated from area volume curve | $A(S_t)$ | $[m^2]$ |
| L | Reservoir losses as evaporation | $A_t * ET$ | $[m^3]$ |
| $Va_t$ | Total available water for allocation without restrictions at the reservoir, during step t | $\max\big(0, S_t + (Q_a(t) + Q_u(t)) * \Delta t - L(t) - V_{buffer}\big)$ | $[m^3]$ |
| $Vr_t$ | Total available water for allocation, from the buffer zone, during step t | $(V_{buffer} - V_{min}) * B_c$: if $Va_t > 0$
$(S_t + Q_a(t) * \Delta t - L(t) - V_{min}) * B_c$: if $Va_t = 0$ | |
| $Vu_t$ | Total available water for allocation during step t | $Vr_t + Va_t$ | |
| $Z_t$ | Reservoir water level at the beginning of step t, calculated from volume elevation curve | $Z(S_t)$ | masl |
| $\Delta Z_t$ | Working water head at step t. | $Z_t - Z_0$ | $[m]$ |
| $H_f$ | Friction losses | $\Delta Z_t * f$ | $[m]$ |
| RPE | Expected generation at step t, expressed as a percentage of installed capacity. | User defined function of local (i.e. reservoir storage at previous timestep, instream flow requirement downstream of turbines, etc) or system-level state variables or context variables (i.e. system level demand, ENSO signal, etc.). See paper Figure 3 | $[\%]$ |
| $V_{pt}$ | Release requirement, to fulfill 100% of expected energy generation at step t, operating at target efficiency | $RPE * P * 1e6 / (9801 * (\Delta Z_t - H_f) * e_{target}) * \Delta t$ | $[m^3]$ |
| $Vd_t$ | Total water available during step t, after the allocation of demands with higher priorities than hydropower | $\max(0, Vu_t - D_t - V_{eco,1})$ | $[m^3]$ |
| $V_{te}$ | Effective turbined volume during step t | $\min(V_{pt}, Vd_t t)$ | $[m^3]$ |
| $Q_{te}$ | Effective turbined flow during step t | $V_{te}/\Delta t$ | $[m^3/s]$ |
| $Vs_t$ | Water available after hydropower is allocated. | $Vd_t - Vt_e$ | $[m^3]$ |
| $V_w$ | Controlled spill volume (floodgates operation) | $V_w = \min\big(R_w^{1/B_w}, Q_w * \Delta t\big)$
with:
$R_w = \max(0, Vs_t - V_{w0})$: Volume above the floodgates operational level | $[m^3]$ |
| $Vs_w$ | Water available after controlled spill is allocated. | $Vs_t - V_w$ | $[m^3]$ |
| $V_{spill_t}$ | Total spill during step t | $\max(0, Vs_w - Vc)$ | $[m^3]$ |
| $P_n$ | Net power output if available flow is turbined given $H_{n,t}$, using n turbines. | $\gamma * \dfrac{Q_{te}}{n} * H_{n,t} * e_{t,n}$ | $[Mw]$ |
| $e_{t,n}$ | Actual Turbine/Generator efficiency given $(\Delta Z_t - H_f)$ and flow $= \frac{Q_{te}}{n}$. See example in | $e_n$
$0$ if $e_n < e_{target}$ | % |

| Value | Description | Calculation | Units |
|---|---|---|---|
| | Figure SI-2. If efficiency is lower than a target efficiency, is adopted as 0. | | |
| $P_t$ | Actual power output for step t | $\max(P_1, 2 * P_2, \dots N * P_N)$ | [Mw] |
| $S_{t+1}$ | Reservoir storage at the beginning of step t | $Vs_t - V_{spill_t}$ | $m^3$ |

[Figure]

**Figure SI-2 Example of a turbine power-efficiency function (Francis type)**

---

## Author Response (AR1)

**AUTHORS' RESPONSE TO EDITOR COMMENTS**

We wish to thank the editor for his thoughtful comments and suggestions to improve the quality and clarity of the manuscript. Individual responses are included below. Changes in the manuscript are highlighted in blue.

**p.7 line 8: what is the unit/dimension of the DOR value? Does it have a physical meaning related to the unit/dimension?**

*AR: We updated section 3.1.2. to clarify the terms and units of eq. 1. See updated manuscript, P.7.*

**p.7 line 12: you write "DORw provides a better estimate…" How do you know? Is there evidence for this?**

We updated section 3.1.2 to explain in more detail the differences between DORw and DOR, and argue that a weighting factor that accounts for basin runoff not affected by artificial regulation can improve the capacity of the index to differentiate the effect of proximity of upstream reservoirs to the reach of interest.

**p.7 line 18: "32-year" or is it 33 –year (see p. 6 line 30, above). Please be consistent.**

We corrected the text. Now reads "33-year"

**p. 14 line 33/p.15 line 1: "As shown, in some cases simple restrictions result in expansion pathways consistently better in most analyzed impacts and benefit metrics." Was this really shown?**

Figure 9 was updated to include boxplots comparing sampling strategies across the basin level metrics. The new graphs show that some sampling strategies perform consistently better.

**p.21 lines17-18: You state: "Our analysis of possible scenarios, however, indicates that other scenarios would result in much lower differential changes." Are there principled lessons to be learned from this – what type of scenarios (i.e. which types of interventions) have lower negative impacts and higher positive impacts?**

We updated the discussion section 4.2 to enumerate the relevant characteristics of hydropower configurations that performed better across the selected metrics.